# Systematic detection of local CH$_4$ anomalies combining satellite measurements and high-resolution forecasts.

Jérôme Barré[1], Ilse Aben[2], Anna Agustí-Panareda[1], Gianpaolo Balsamo[1], Nicolas Bousserez[1], Peter Dueben[1], Richard Engelen[1], Antje Inness[1], Alba Lorente[2], Joe McNorton[1], Vincent-Henri Peuch[1], Gabor Radnoti[1], Roberto Ribas[1]

[1]ECMWF, European Centre for Medium Range Weather Forecasts, Shinfield Park, Reading, United Kingdom
[2]SRON, Netherlands Institute for Space Research, Utrecht, Netherlands

*Correspondence to*: Jérôme Barré (jerome.barre@ecmwf.int)

**Abstract.** In this study we present a novel monitoring methodology combining satellite retrievals and forecasts to detect local CH$_4$ concentration anomalies worldwide that are related to rapidly changing anthropogenic emissions that significantly contribute to the CH$_4$ atmospheric budget but also that are related to biases in the satellite retrieval data. The method uses high resolution (7 km x 7 km) retrievals of total column CH$_4$ from the Tropospheric Monitoring Instrument (TROPOMI) onboard the Sentinel 5 Precursor satellite. Observations are combined with high resolution CH$_4$ forecasts (~9 km) produced by the Copernicus Atmosphere Monitoring Service (CAMS) to provide departures (observations minus forecasts) close to the native satellite resolution at appropriate time. Investigating the departures is an effective way to link satellite measurements and emission inventory data in a quantitative manner. We perform filtering on the departures to remove the synoptic-scale and meso-alpha-scale biases on both forecasts and satellite observations. We then use a simple classification on the filtered departures to detect anomalies and plumes coming from CAMS emissions that are missing (e.g., pipeline or facility leaks), under-reported or over-reported (e.g., depleted drilling fields). The classification method also shows some limitations to solely focus on emissions detection due to local satellite retrieval biases linked with albedo and scattering issues.

## 1. Introduction

Atmospheric methane (CH$_4$) is the second most important anthropogenic greenhouse gas after carbon dioxide and contributes significantly to changes in radiative forcing and climate change. CH$_4$ is estimated to account for at least a quarter of the present-day warming (Myhre et al., 2013) and has a near-term global warming potential that is 84 times larger than CO$_2$ per unit mass (IPCC 2013). There are numerous natural and anthropogenic CH$_4$ sources, which vary in location and areal extent. The anthropogenic emissions related such as oil and gas production and coal mining and biomass burning tends to be geographically localised, e.g., over a plant facility, a pipeline or a field of extraction. Methane emissions however related to

biological fluxes such as livestock, landfills and rice fields which can also be either geographically localised over narrow areas or more widespread. For example, microbial respiration in wetlands showing more extensive patterns over the globe (Saunois et al., 2016). Atmospheric methane concentrations have more than doubled since the pre-industrial times because of the imbalance between methane sources and sinks (IPCC, 2013), due to an increase of oil and gas production, rice crops, livestock and landfills. Methane has a relatively short atmospheric lifetime (with respect to climate scales) of around 9 years, meaning

targeted emission reductions could be an effective way to limit the rate of warming over the upcoming decades (Shoemaker et al., 2013).

Greenhouse gases emission inventories are generated using aggregation and extrapolation of regional and national specific data. These data are reported individually by countries using the guidelines provided by the United Nations Framework on Climate Change (UNFCC) and the Intergovernmental Panel for Climate Change (IPCC). The reporting follows a bottom-up

approach, which utilises activity data and emission factors of individual emissions sectors. Official reporting and processing of this data to build these bottom-up inventories can cause significant lag and information can be out of date for certain sectors once publicly released. This can become an issue in the context of rapidly changing emissions from large point sources, for example in the oil and gas sectors (Alvarez et al., 2018). In the case of atmospheric composition modelling, emission inventories are used for input surface fluxes to simulate atmospheric concentrations. Within the Copernicus Atmosphere

Monitoring Service (CAMS) these simulations are used to provide routine real-time forecasts of greenhouse gases concentrations. The CAMS greenhouse gas forecasting system integrates satellite observations (Massart et al., 2014, 2016) to generate initial conditions for high-resolution forecasts at about 10 km (Agustí-Panareda et al., 2019). The lack of up-to-date emission inventories will impact and likely degrade simulated $CH_4$ concentrations in areas where the local contribution of anthropogenic emissions is significant.

Many studies have shown the rapidly changing and event-based nature of $CH_4$ anthropogenic emissions, especially in the case of identifying the location of 'super-emitter' point source locations. Conley et al. (2016) used aircraft measurements to characterise a blowout of a well connected to the Aliso Canyon gas storage facility in California from October 2015 to February 2016. Pandey et al. (2019) showcased detection of large methane emission from a gas well blowout in Ohio during February to March 2018 using satellite measurements. More recently, Varon et al. (2019) detected an anomalously large $CH_4$ source

using a combination of satellite instruments over Central Asia (western Turkmenistan) associated with a gas compression station. Those types of suddenly occurring $CH_4$ emissions cannot be or are not reported/detected in time to be included in the bottom-up inventories but are seen from space. Other studies showed the capability of satellite measurements to detect $CH_4$ emissions related to extensive drilling and fracking areas. Kort et al. (2014) identified a large methane anomaly over the Four Corners region of the USA and more recently de Gouw et al. (2020), Zhang et al. (2020) and Schneising et al. (2020) showed

satellite detection of large and extended enhancements in different US oil and gas production regions such as the Permian basin. While these satellite-based studies focused on specific events and locations, none of them systematically detected such anomalies at global scale, nor did they provide a method to do so.

Systematic detection of large point sources of anthropogenic CH4 emissions using a combination of satellite observations and modelling could enable rapid action to reduce emissions from the oil and gas sectors. Two recent developments allow for systematic detection of unreported CH4 atmospheric anomalies linked to small scale and point sources emissions. Firstly, newly available high resolution (7 km x 7 km) satellite observations from the TROPOspheric Monitoring Instrument (TROPOMI, Veefkind et al., 2012) on board the Sentinel-5p platform. Secondly, improved real-time forecasting at high-resolution (~9 km) provided by CAMS (Agustí-Panareda et al., 2019). In this paper we present a novel methodology to routinely compare the satellite observations with the model forecasts in order to perform a systematic detection of atmospheric CH4 anomalies related to emission changes from small scale and point sources emissions that are not reported or lack timely update. The paper is organised as follows: Section 2 describes the setup that includes the TROPOMI observations, the forecasting and monitoring configurations, Section 3 presents the detection method, Section 4 discusses several case studies to showcase the capabilities but also the limitations of the detection method. This is followed by conclusions where we discuss briefly the benefit of our approach with coarse resolution inverse modelling.

## 2. Setup

### 2.1. TROPOMI CH4 observations

The TROPOMI (Veefkind et al., 2012) instrument was launched 13 October 2017 onboard the Sentinel-5 Precursor satellite, a low Earth orbiter with a Sun-synchronous orbit that overpasses at 13:30 local solar time. Currently operational since the end of April 2018, the instrument is an imaging spectrometer with a wide spectral range: ultraviolet, visible, near infrared and short-wave infrared. This allows TROPOMI to measure a variety of atmospheric chemical species such as: ozone, nitrogen dioxide, carbon monoxide, sulphur dioxide, formaldehyde, aerosol and methane (Hu et al., 2018). Current CH4 observations, which are available for the inner two thirds of the swath and only over land, are vertically integrated columns sensitive to the troposphere (surface to 200 hPa). With a swath of around 1,750 km (normally 2,600 km) wide from the along track position and a ground pixel size of 7 km x 7 km, TROPOMI CH4 data can provide near global daily coverage at high horizontal resolution over land but is limited by cloud cover and retrieval quality. In this study, we use the bias corrected version of the product and we apply the most stringent quality flagging possible, selecting only pixels that have the *qa_value = 1.0* (see Product Readme Methane V01.03.02, https://sentinel.esa.int/documents/247904/3541451/Sentinel-5P-Methane-Product-Readme-File). In the later document it is stated that an overall bias of -0.3% is found with comparison against independent data and is well within the mission requirements of ≤ 1.5% (24 ppb). The scatter of the data around this bias also complies with mission requirements of ≤1.0% (18 ppb). Figure 1 illustrates the CH4 satellite observation coverage that TROPOMI provides over a year, a month and a day.

The measurements show clear geographical variation of the CH4 column-averaged dry-air mixing ratios (XCH4) that are driven by the atmospheric transport but most importantly by the spatial and temporal variability of the surface fluxes and emissions variations. Figure 2 shows the 2019 annual average zoomed over the Middle East region and the western USA

regions. Over these regions, spatial variability results in XCH4 enhancements of up to 50 ppb over emission hotspots. Differences in the average concentrations from region to region are also significant, from approximately 1825 ppb over the USA to 1875 ppb over the Middle East. The strong local enhancements are an indication of strong local surface fluxes and emissions of CH4 from oil and gas activities, mining, agriculture or wetlands. XCH4 retrievals can also be prone to some systematic residual errors especially related to surface albedo (Hasekamp et al., 2019). De Gouw et al. (2020) for instance mentioned the possibility of retrieval biases due to low surface albedo in the short-wave infrared spectral bands in the winter. Such retrieval biases, even though mostly reduced by the bias corrected product, need further investigation (see section 4.3). Nevertheless, the TROPOMI data are sufficiently accurate to show local enhancements linked (but not limited) to oil and gas production. We show in section 3 how to isolate these small-scale signals of interest and how to remove the contribution of synoptic-scale (more than 2000 km) and meso-alpha-scale (between 2000 km and 200 km) biases. In the rest of the paper, we define large-scale as the combination of synoptic-scale and meso-alpha-scale.

### 2.2. CAMS high-resolution CH4 forecasting suite

In this study we use the ECMWF Integrated Forecasting System (IFS), which is used in different configurations for the operational Numerical Weather Prediction (NWP) system as well as for the Copernicus Atmosphere Monitoring Service (CAMS) atmospheric composition analyses and forecasts. As part of the CAMS greenhouse gases services, the IFS is used to provide 5 days $CO_2$ and $CH_4$ forecasts (Agustí-Panareda et al., 2019) jointly with other species relevant for air-quality (Flemming et al., 2015).

The IFS model cycle used in this paper is CY45R1 and is run routinely with a TCo1279 horizontal resolution which is a cubic octahedral reduced Gaussian grid at approximately 9 km (Holm et al., 2016) with 137 vertical levels from the surface to 0.01hPa and a time step of 450 seconds. Details about the transport and meteorological configuration can be found in Agustí-Panareda et al. (2019). The CAMS greenhouse gases (GHG) operational suite is composed of an analysis and forecasts at medium and high resolution (see Fig. 3). The analysis is based on the IFS 4D-Var assimilation system which was adapted to assimilate retrieved columns-averaged mole fractions of $CO_2$ and $CH_4$ together with all the operational meteorological observations (Engelen et al., 2009, Massart et al., 2014, 2016). The analyses are produced every 12hours (00:00UTC and 12:00UTC). A 4-day forecast is then issued daily after the 00:00UTC analysis on a TCo399, a cubic octahedral grid corresponding to approximately 25 km x 25 km with the same 137 model level configuration. Two satellite observation streams are currently assimilated, the Infrared Atmospheric Sounding Interferometer (IASI) for $CH_4$ on the MetOp satellites and the Thermal And Near-infrared Sensor for carbon Observations (TANSO) on the GOSAT satellite for both $CO_2$ and $CH_4$ (see Massart et al. (2014) for further details). In this configuration only the concentrations are corrected by the assimilation, the emissions and surface fluxes remain unchanged. The processing and acquisition of the level 2 data in 2019 provide the satellite XCH4 data 4 days behind real time. The high-resolution forecast is then coupled to the analysis experiment by merging the 4-day lower resolution forecast from the $CO_2$ and $CH_4$ analysis with the previous 1-day high resolution forecast (Fig. 3) in order

to preserve the fine-scale features of the high-resolution forecast. Additionally, the high-resolution forecast coming from the operational NWP runs is used to reset the initial meteorological conditions in order to ensure the best possible accuracy of the transport. In this paper we will focus on using the $CH_4$ forecasts at high-resolution coming from the setup described above.

The high-resolution forecasts are run on a TCo1279 L137 grid of approximately 9 km x 9 km for a 5-day period and are initialized approximately 4 hours behind real-time every day from 00:00UTC. The impact of the TANSO and IASI $CH_4$ retrievals assimilation is not strong close to the surface (see Fig. 8 in Massart et al., 2014). Analyses performed at 25km of resolution are not correcting the emissions and hence will mainly provide a correction in the forecast initial condition concentrations in the free troposphere and above. At lower altitudes the influence of the emissions in the forecast is dominating

during the 4-day forecast at 25 km that is used to initialise the high-resolution forecast at 9 km that does not include data assimilation.

Both high resolution forecasts and analysis use prescribed $CH_4$ surface fluxes. The anthropogenic emissions including fossil fuel emissions, agriculture and landfill/waste emissions are from the annual EDGARv4.2FT2010 data set (Olivier and G. Janssens-Maenhout, 2012) for 2010 with 0.1°x0.1° resolution and monthly resolution for the rice emissions (Matthews et

al., 1991). Monthly mean wetland emissions come from a climatology (1990-2008) based on the LPJ-WHyMe model constrained by SCIAMACHY observations during the HYMN project (Spahni et al., 2011) with a resolution of 1° x 1° degree. The biomass burning emissions are from GFASv1.2 (Kaiser et al., 2012). Other sources and sinks include a monthly soil sink (Ridgwell et al., 1999), annual mean oceanic fluxes (Houweling et al., 1999, Lambert and Schmidt,1993), and monthly mean fluxes from termites (Sanderson, 1996) and wild animals (Houweling et al., 1999). The chemical sink in the troposphere and

the stratosphere is represented by a climatological monthly mean chemical loss rate (Bergamaschi et al., 2009). This is based on OH fields optimised with methyl chloroform using the TM5 model (Krol et al., 2005) with prescribed concentrations of the stratospheric radicals using the 2-D photochemical Max Planck Institute model. Figure 4 shows the geographical and seasonal structure of the surface fluxes. Large-scale and smoother structures are representative of the wetland, soil and agriculture fluxes, whereas the finer-scale and shaper structures are representative of the anthropogenic and fire emissions. Figure 5 shows

the capability of the high-resolution forecasts at global and regional scales. Global seasonal cycles and synoptic scale concentrations are represented as well as concentrations at smaller scales such as plumes from point source emissions and orographic effects. Large point sources and associated plumes can be seen over Europe, for example over Madrid, Paris and Tours (western France). Inventory estimates suggest the modelled hotspot region near Tours is probably the result of solid waste landfill emissions. Other possibilities include emissions from both the enteric fermentation and wastewater treatment

sectors, all of which may be linked to a landfill site. Over the Middle East region zoom, sharp point sources are seen in Teheran and Southern Iran as well as over Pakistan (Karachi) and also closer to the Himalayan region.

### 2.3. Monitoring suite

To monitor and compare the TROPOMI XCH$_4$ retrievals with the IFS CH$_4$ 9 km forecasts we re-use a part of the IFS assimilation system in a so-called monitoring mode. The system recomputes a high-resolution trajectory at 9 km (which is a model integration) initialised from the forecasts over a 12-hour monitoring window to calculate so-called first guess departures (difference between the observation and the model forecast) with the observations at the appropriate model time step. At each observation location the departure can be written as follows,

$$d = y - \boldsymbol{H}\boldsymbol{M}(x_i) \ (1)$$

where $d$ is the departure, $y$ the observation, $\boldsymbol{H}$ the observation operator, $\boldsymbol{M}$ the model integration or trajectory and $x_i$ the initial CH$_4$ condition at the beginning of the monitoring window. If we inject the retrieval equation (Rodgers, 2000) the departure becomes,

$$d = \boldsymbol{A}x_t + (\boldsymbol{I} - \boldsymbol{A})x_a + \epsilon - \boldsymbol{A}\boldsymbol{M}(x_i) - (\boldsymbol{I} - \boldsymbol{A})x_a \qquad (2)$$

where $x_t$ is the true CH$_4$ concentration state (which is never exactly known), $\boldsymbol{A}$ is the averaging kernel matrix which represents the sensitivity of the retrieval on the vertical profile with respect to the true state, $\boldsymbol{I}$ the identity matrix, $x_a$ the apriori information used in the retrieval and $\epsilon$ the retrieval error term. The equation then simplifies to,

$$d = \boldsymbol{A}\big(x_t - \boldsymbol{M}(x_i)\big) + \epsilon \qquad (3)$$

which is the difference between the true state and the forecast smoothed by the averaging kernel function plus the retrieval error term. Those departure values are thus strongly dependent on the averaging kernel function shape. For the TROPOMI XCH$_4$ retrievals the mean averaging kernel function shows a homogenous sensitivity to the entire atmosphere where the sensitivity slightly decreases in the stratosphere (see Fig. 2 in Hu et al., 2016). The averaging kernel function is not very variable between pixels or between different regions of the globe (not shown). Figure 6 shows the departures over various time scales (yearly, monthly and daily) for the global domain. Overall, the departures (observation minus forecast) show a global positive bias of around 25 ppb (meaning observation values are above the model values) which could be attributed to model biases (Ramonet et al., 2019) and/or observation biases (Langerock et al., 2019). Ramonet et al. (2019) compared the CAMS CH$_4$ forecasts with independent total column data. Results showed that the forecasts continuously underestimate the CH$_4$ total columns by 5-20 ppb. Langerock et al. (2019) showed that the averaged total column bias for the TROPOMI CH$_4$ retrievals bias is -0.32% (i.e., around -5ppb) but with respect to ground-based measurements.

Regional-scale error structures are evident from the observation-model comparison. For example, boreal regions are showing a band of negatives values, potentially attributed to systematic errors caused by surface albedo values during winter (see section 2.1) in the TROPOMI retrieval algorithm. Alternatively, they could be caused by CH$_4$ biases at tropopause and lower stratosphere levels in the IFS model. Also, a possible time lag in the wetland emissions, which are calculated offline and provide boundary conditions in the IFS forecasting chain (see section 2.2) could cause such bias. The attribution of this type of large-scale error seen in the departures in not fully understood yet and is beyond the scope of this paper, although an understanding of these biases is crucial to further improve the quality of the CAMS CH$_4$ forecasts and TROPOMI retrievals.

At finer scales, structures are seen on the yearly average comparison and become more evident on the monthly timescales. Local differences are even stronger on a daily basis but recognising fine scale structures is challenging due to the lack of daily coverage. For those reasons a spatial filtering and temporal averaging of the departures is performed to extract and use the small-scale features seen in the departures.

## 3.    Detection method

### 3.1. Filtering the signal

To remove the large-scale features seen in the departures we have implemented a high pass Gaussian filtering. The filter uses a convolution of a 2D Gaussian kernel on a given averaged and binned departure field. In this study we use a $0.1°$ latitude-longitude binning. Due to ocean, cloud cover and quality control flagging a number of bins of the departure will show missing values that will jeopardize the convolution. This problem is solved technically by creating two auxiliary matrices that have
missing values replaced by 0. The two auxiliary matrices are then defined as

$$\boldsymbol{D} = \begin{cases} d_m, & if \ n > N \\ 0, & otherwise \end{cases} \tag{4}$$

$$\boldsymbol{C} = \begin{cases} 1, & if \ n > N \\ 0, & otherwise \end{cases} \tag{5}$$

where $d_m$ (with the subscript $m$ standing for mean) is the average departure in the given bin, $N$ the threshold of minimum number of observations in a given bin. In this study, we have chosen $N = 2$ in order to avoid smoothing with very isolated pixels that can be faulty but also keep as much data as possible. Replacing the missing values by zeros in $\boldsymbol{D}$ introduces an error after convolving (inducing low values due to smoothing out with zeros) in the filtered departures $d_{hp}$ (with the subscript $hp$ standing for high pass). This can be compensated by applying the same Gaussian filter on a matrix $\boldsymbol{C}$ representing the selected
bins for filtering (where number of counts are above $N$) and using the ratio of the two filtered matrices to compensate for the missing value errors. Then a high pass filtering on a given observation space field (here departures) can be formulated as follows,

$$d_{hp} = d_m - \frac{G(\sigma) * \boldsymbol{D}}{G(\sigma) * \boldsymbol{C}} \tag{6}$$

where $G(\sigma)$ is a 2D Gaussian kernel function with a $\sigma$ length scale. The same filtering is also applied on the observation
values $y$ and the first guess values $\boldsymbol{HM}(x_b)$ as this will be used for classification in section 3.2. Figure 7 shows the effect of the filtering on the observation-space data using a 30-day window and a length-scale of $2°$. Firstly, we can see that the large-scale features in the departures such as the overall bias and regional variations are removed. Secondly, the departures,

observations and first guess distributions are put towards gaussianity, centred around zero and displaying more a symmetrical shape and tails. This then makes the processing and the classification of the data much easier (see section 3.2).

To decide on the appropriate window length and Gaussian kernel length scale we have conducted sensitivity tests with different length scales ($\sigma$= [0.5,1.0,2.0,5.0] degrees) and a window length of 10, 30 and 90 days. Figure 8 shows the resulting filtered departures normalized by the instrument precision for the 12 possible sensitivity tests. For tests with Gaussian kernel sizes of 0.5 and 1.0 degree the filtered signal is mostly weaker than the measurement precision and very few to no detections of local anomalies will be made. Conversely, if the kernel is large the relative signal over the instrument precision

is stronger but to the risk of picking up larger patterns than the targeted features, i.e., features that are directly related to local emissions in the $CH_4$ atmospheric distribution. For these reasons we found that a kernel of 2.0 degrees performs best. If the time window is short, e.g., 10 days, lower coverage could limit a correct detection of outliers especially in the case of isolated data points. Isolated data points that are spotting possible outliers could be filtered out towards 0 as the convolution do not have neighbouring points to use within the kernel range. The shorter the window and the narrower the kernel, the more likely

this can occur. Conversely, if the time window is long, i.e., 90 days, this will maximise the chances to have a good observation coverage for the convolution filter to run best. But this would reduce the ability to provide information on temporal variability. Also, the sharp spatial structures that correspond to more recent or sporadic emission events are smoothed in the time averaging effect decreasing the filtered departure over instrument precision ratio. For those reasons, we found that a time window of 30 days provides the most reasonable results.

## 3.2. Outlier classification


The final step is an outlier detection of the filtered departures. We choose to retain the values which have a filtered departure absolute value superior to the TROPOMI $CH_4$ measurement precision. If the filtered departures absolute values are lower than the measurement precision, they are then considered as noise and ruled out. Further refinements to the current methodology could be done to find more optimal method for outlier detection using more advanced statistical methodologies. In the present

study we found that the provided measurement precision with the satellite product provides suitable results. In addition to the outlier detection, we perform a classification given the relative values and sign of the filtered observations and forecast values. This allows us to define the following four categories:

- **high observations (red in Fig. 9):** where filtered observations values are higher than filtered forecast absolute values. This class is representative of high XCH4 values detected by TROPOMI that are not seen as high or at all in the forecasts.

These are likely originating from emissions that are not reported or under-estimated in the inventories. However, high observation categorisation may also be caused by poor quality observations due to albedo and scattering issues (see section 4.3).

- **high forecasts (green in Fig. 9):** where filtered forecast values are higher than filtered observations absolute values. This class is representative of high $CH_4$ values in the forecasts but not seen as strong or at all in the TROPOMI XCH4 retrievals.

High forecasts categorized data points are likely originating from emissions that are over-estimated or no longer being produced or even mis-located in the emission inventory.

- **low observations (blue in Fig. 9):** where filtered observations values are lower than filtered forecast absolute values. This class is representative of locally low $XCH_4$ values detected by TROPOMI but are not seen to be as low or at all in the forecasts. Poor-quality observations influenced by low surface albedo likely fall in that category (see section 4.3).

- **low forecasts (gold in Fig. 9):** where filtered forecast values are lower than filtered observations absolute values. This class is representative of low $XCH_4$ values in the forecasts but not seen as low or at all in the TROPOMI $XCH_4$ retrievals. This category has generally much fewer data points very sparsely distributed. Orography could be a reason for data points to fall in that category, i.e., model surface height value that are higher than the observation value. Further developments of the method will likely use orography to improve the filtering.

In the maps of Fig 9., shades of the colours indicate the intensity of the offset, i.e., how far from perfectly matching observation versus forecasts the filtered departure is. The size of the points indicates the number of samples. A larger dot indicates more data points within the 30-day window to compute the statistics hence is more robust. Fig 9. gives an overview of such detections globally and cases are many and various. In the next section we will focus on specific cases studies using the under-reported or missed plumes (red) category and the over-reported plumes (green) category to showcase the usefulness
of the method but also its current limitations.

## 4. Case studies

### 4.1. Under-estimation of local sources in the forecasts

**South Western USA and Mexico:** In figure 10, the method detects under-predicted local $CH_4$ concentrations (in red) in the forecast system in three areas. This occurs in the Permian Basin region, located around the Texas-New Mexico border,
where multiple oil drilling sites are currently operating. Those enhancements have been documented by de Gouw et al. (2020) and Zhang et al, (2020) showing the reliability of the presented method. Two other regions with a smaller bias and extent can be identified around the southern tip of Nevada by lake Meade and northern Baja California close to the US-Mexican border. To our knowledge those two cases have not been investigated or documented yet and need further investigation. We did not identify a facility responsible for those enhancements and we also did not find matching land surface albedo features that could
create local biases in the retrievals (see section 4.3) in visible satellite imagery. In the case of the northern Baja California enhancements correlations of the enhancement with scattering parameters such as the aerosol optical thickness (AOT) are seen (not shown) and needs to be considered for further investigation to improve the method. A case study of the influence of the AOT on the detection method is described in section 4.3.

**Western Turkmenistan:** To confirm the ability of this methodology for the detection of large point-source emitters we
also showcase very strong detection of anomalous concentrations over the western Turkmenistan. Our system detects strong

enhancement during most of 2019 (Fig. 11) that change in intensity and shape. The filtered departures can be very large (above 50 ppb) with a high number count in the bins (large size of the dots). As mentioned earlier in this paper, anomalously large $CH_4$ sources from oil and gas production in this location have been documented and detected using TROPOMI combined with private sector satellite data by Varon et al. (2019).

## 4.2. Over estimation of local sources in the forecasts

**Western Russia:** Our detection system shows two local point sources, that show large forecast values that are not seen by TROPOMI $XCH_4$ (green dots in Fig. 12). The features do not show large sampling (small dots) in time but do exhibit the shape of plumes, with strong departures near the point sources. One is very close to Moscow and corresponds to the Domodedovo airport surroundings. The other source detection is near the Volga river with a location matching small drilling fields seen in visible satellite images. In these two locations the detection method suggests that emission inventories are over-estimating local sources, which in reality are now producing reduced emissions or are no longer active emitters (at least during the period of monitoring).

**Los Angeles**: Similar features can occur in the area of Los Angeles. Figure 10 shows significant over-prediction of $CH_4$ (green dots) specifically over San Bernardino and Palmdale. Both towns have industrial facilities and regional airports. The detection is stronger in the 2019-09-01 and 2019-10-01 windows than in the 2019-07-01 and 2019-08-01 windows. Differences in intensity could be attributed to the monthly emission changes but also attributed to seasonal atmospheric transport changes due to different meteorological situation between windows. For example, if the overall windspeed increases near the source less accumulation of $CH_4$ would be seen leading to smaller departures and less detection.

Such cases in very different locations show the capability of the method to detect not only missing or underreported point sources but also overreported cases. This can only be achieved with combining numerical models forecasts and satellite measurements at close-matching high horizontal resolution (9 km and 7 km respectively). It is also important to mention that the method presented here is subject to uncertainties due to both model transport errors and representation error, although the error associated to emission generally dominate. Further work is needed to account for atmospheric transport and more generally to account for the weather variability in the detection method. Techniques as described in Barré et al., 2020 show interesting potential to be used for this topic.

## 4.3. Local retrieval issues

The TROPOMI $XCH_4$ retrieval can be affected by albedo issues (see section 2) and the filtering is not able to remove features with geographical extent smaller than the size of the Gaussian kernel (see section 3.1). Figure 13 shows patterns in the outlier detections where filtered observations values are lower than filtered forecast (low observations category in blue). Similar patterns are repeated over two months. In figure 14 the TROPOMI albedo in the near infrared (NIR) and the short-wave infrared (SWIR) bands are displayed with the $XCH_4$ column retrieval. Variations of the albedo in the two spectral bands are strong in this area and this affects the $XCH_4$ columns. The patterns between the albedo and the $XCH_4$ columns are clearly

matching. Depending on the structure of the pattern and how narrow or small it is, the filtering algorithm will not be able to remove this effect. This is currently affecting the outlier detection method.

Another example is provided over the Siberian region in figure 15 where the same pattern is seen repeatedly in the outlier detection. In this case, the filtered observations values are higher than filtered forecast (high observation category in red). The patterns detected are matching with the variations of the SWIR albedo in figure 16. Here the features seen in the SWIR albedo produces consistently higher TROPOMI XCH$_4$ values than its surroundings and the patterns are narrow enough to be missed by the filter. Thus, great care should be taken when diagnosing such filtered departures. Features with a consistent distinctive

shape and intensity are potential retrieval error artefacts as in most cases atmospheric methane signals would show more variability as a result of meteorological dynamics and not a consistently distinctive shape over months (8 and 10 weeks in Fig. 13 and Fig. 15, respectively). In the further evolution of our detection method, albedo information should be included to try to account for such systematic local biases.

       However, in certain cases, the identification of such biases can be more complex than just correcting outlier detections

using albedo information. In Fig. 17 the shape of the anomaly is also consistent over 4 months using a 30-days windows but no clear correlated feature in the NIR and SWIR albedo is standing out in that case. Investigations showed that this feature is associated with scattering parameters such as the aerosol optical thickness (AOT). In this case the feature is correlating well with both NIR and SWIR bands of the AOT. As an example, Figure 18 shows the presence of the AOT feature in the NIR band that correlates to the detection high XCH$_4$ observation against the forecasts. Similar feature is seen in the SWIR band

AOT as well (not shown). Such issue is showing that more work is needed  identify systematic local biases that are not only caused by albedo variations. Additionally, there is still potential impact of temporally and spatially variable small-scale albedo features. For example, not temporally persistent biases can arise with snow coverage. More research will be needed to consider an automatic local biases detection for less persistent biases issues.

       Currently the method presented here helps identifying retrieval biases but do not systematically do so. Further

improvements of the method could be implemented in the future using jointly albedo and scattering parameters to perform additional filtering correction and flagging. Having such retrieval bias automatic detection feature available would help improving the quality of the retrieval product.

**Conclusions**

       In this paper we have shown the potential of systematic detection of anthropogenic CH$_4$ point and local source emissions

relative to known emission inventory data using the TROPOMI satellite measurements in combination with high resolution CH$_4$ forecasts. While many studies have shown detailed analysis of a few case studies using TROPOMI observations, this is the first step to provide a systematic way to detect strong anthropogenic local emitters of CH$_4$ and to compare results with emission inventories. The method presented here does not only show the potential for detection of unreported or missing sources but also targets over-reported sources in the inventories. The method also has the potential of identifying systematic

local retrieval errors which could help to improve the satellite product in return. The current method however has some limitations as it requires additional correlative analysis with albedo and scattering parameters to account for local biases in the retrieval values. Complications could arise if land surface induced biases and local emission detected features would be collocated. The case of multiple type of emissions collocated (i.e., anthropogenic plus wetland emissions) will also further increase in complexity as emission patterns could show albedo correlations. We however have demonstrated the potential of

the methodology by focusing on several case studies and further work is required to provide a global assessment using several years from this dataset.

Our method is novel by combining information from multiple sources (emission inventory, modelled surface fluxes, and observations) in a data assimilation framework to detect and analyse observed anomalies. We have used global emission inventories and fluxes that were the best possible global estimates we had available at the time when running our system. Using

different emission inventories from research specific activities that are more specific to local regions, for instance, could provide different answers. In that way our methodology could provide an efficient way to validate improvements in sector-specific emission inventories. For example, using revised $CH_4$ inventories such as presented by Maasakers et al. (2016) over the USA or more recently by Scrapelli et al., 2020 globally could lead to different detection patterns. Bottom-up inventories will always lag in time and therefore cannot track rapid emission changes such as pipeline and gas facility blowouts. Satellite

measurements have a clear added value for timely detection in the case of large emissions.

Combining satellite measurements, forecasts and emission inventories partially using a data assimilation system paves the way to estimate the emissions themselves. Inverse modelling studies to estimate $CH_4$ emissions have been done with SCIAMACHY and GOSAT $CH_4$ satellite data generally performed at rather low resolution and focus specific study sites (e.g., Jacob et al., 2016). To our knowledge no published studies showed global inversions using TROPOMI data updating emissions

close to the 10 km scale globally. Inverse modelling is computationally expensive and in the case of running operations beyond 10 km scales to close-match satellite observations is a challenge that needs to be overcome over the next decade. Efforts are underway to implement a sector-specific inverse high-resolution modelling monitoring system as part of the CAMS service evolution at ECMWF and the future Copernicus CO2 service at global and regional scales (e.g., Barré et al., 2019, Bousserez et al., 2019, Pinty et al., 2019, Janssens-Maenhout et al., 2020). Approaches combining global and regional modelling could

be adopted to perform inversion at fine scales but at the cost of missing fine-scale detection outside the regional domains. Large and local $CH_4$ emissions events could occur in very remote areas, which are typically not considered in regional modelling setups (e.g., West Turkmenistan, Varon et al.,2019). Systematic detection will then require setting up many regional subdomains leading again to computational burden for a single monitoring entity. We have demonstrated that monitoring of satellite $XCH_4$ departures at high resolution at global scale using already existing parts of a forecasting chain

remains an affordable solution to develop a much-needed capability: tracking rapidly changing $CH_4$ sources across the world and support the urgently needed effort on developing climate policies for reducing anthropogenic $CH_4$ emissions.

**Author contribution**

J. Barré prepared the manuscript with the contribution of all co-authors. J. Barré designed the experiments and the system from G. Radnoti and A. Agustí-Panareda developments. I. Aben and A. Lorente helped with satellite data expertise.

**Competing interests**

The authors declare that they have no conflict of interest.

**Acknowledgements**

This work was produced by the Copernicus Atmosphere Monitoring Service (CAMS) which is implemented by the European Centre for Medium-Range Weather Forecasts (ECMWF) on behalf of the European Commission. We thank the two anonymous reviewers for their helpful comments and suggestions that improved this paper.

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

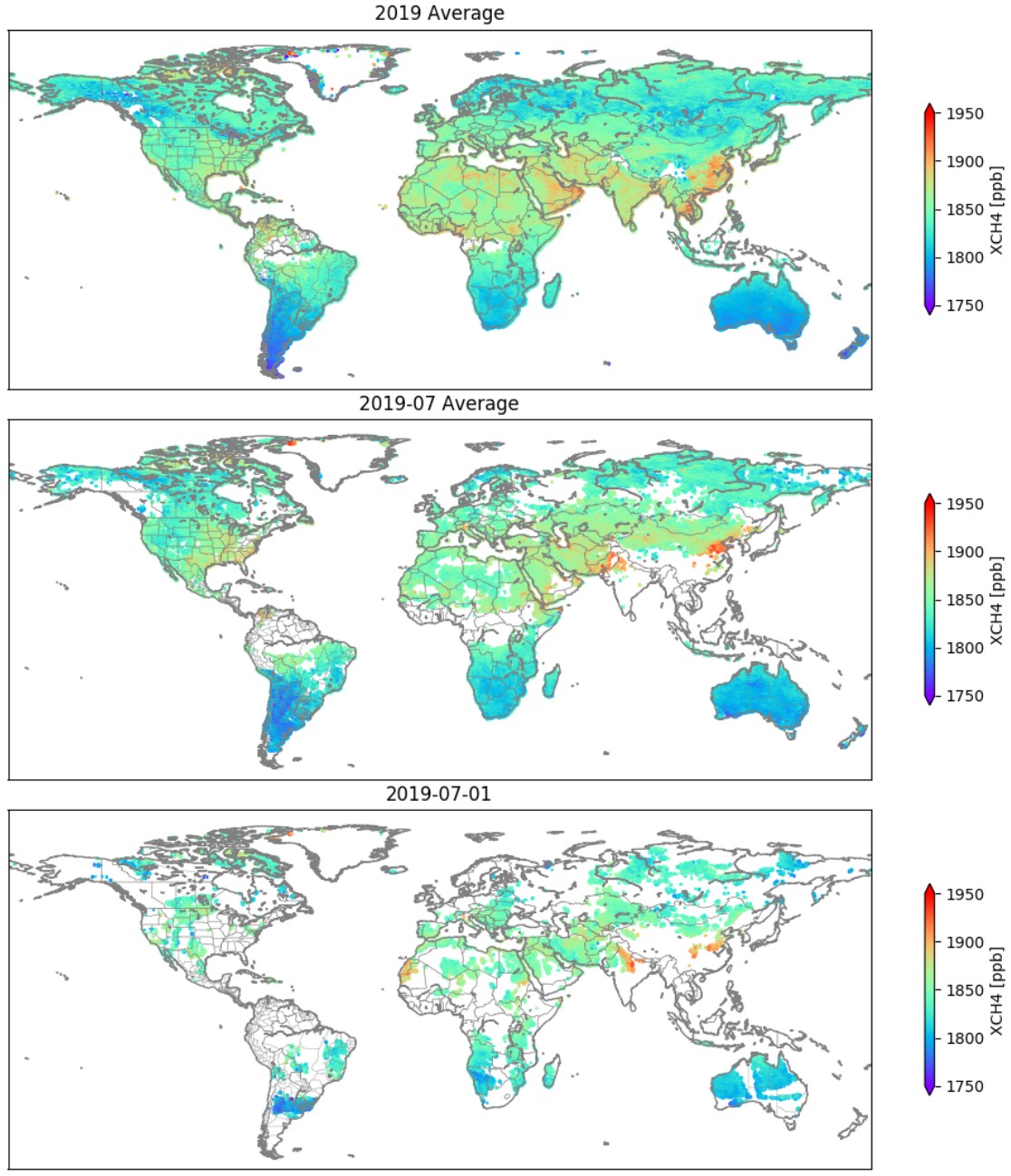

**Figure 1. Global average of TROPOMI XCH4 column-averaged dry-air mixing ratios for the full year 2019, July 2019 and July 1st, 2019 (top to bottom).**


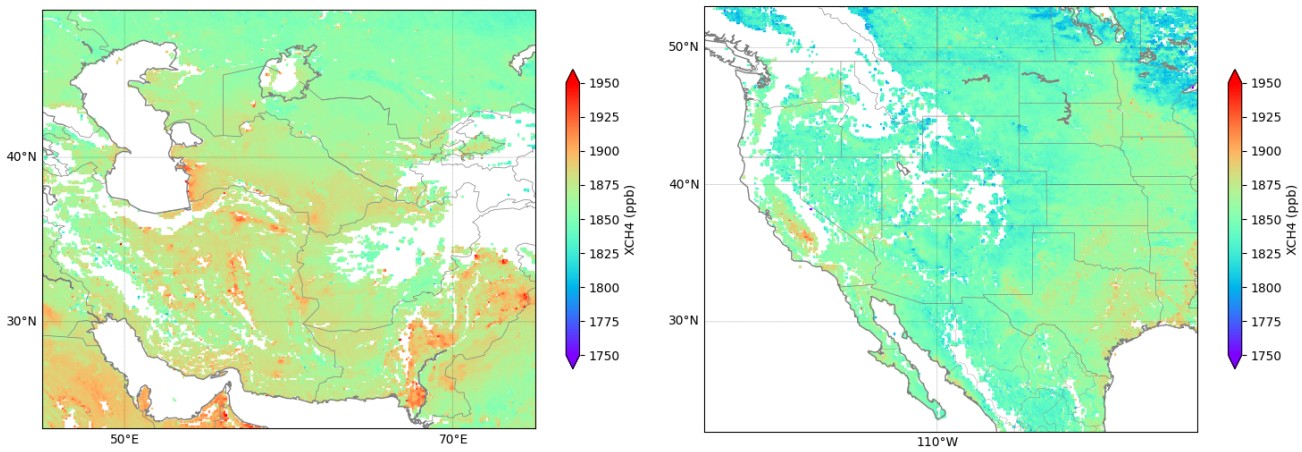

570 **Fig 2. Regional zooms of TROPOMI XCH₄ columns for the full year 2019. Middle East (left) and Central-Western North America (right).**

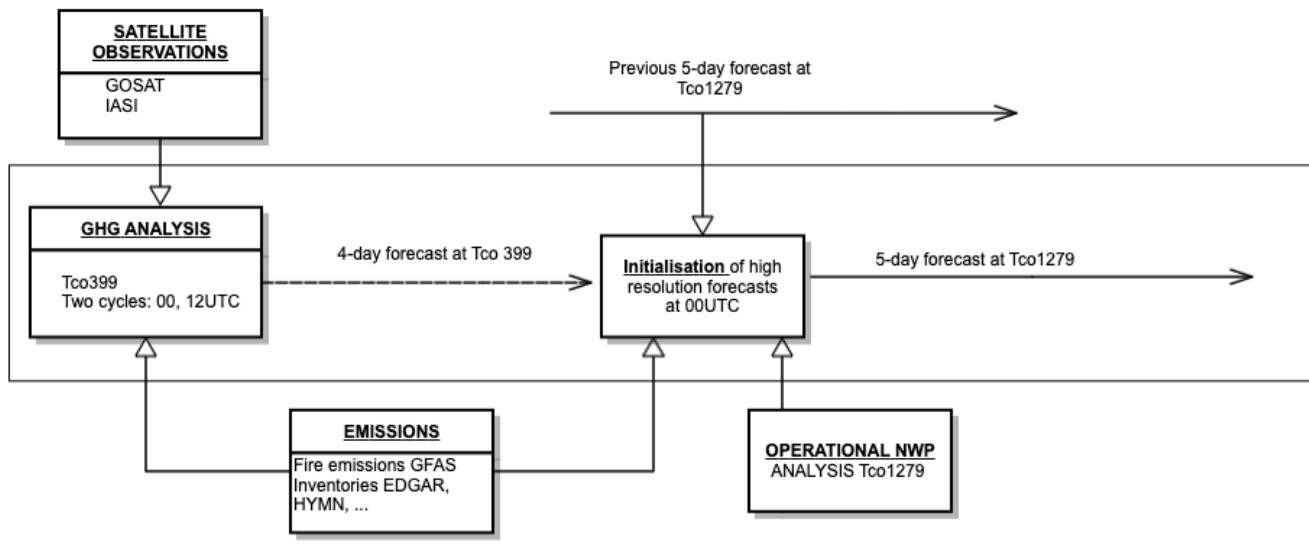

575      **Figure 3. Flow chart of the CAMS greenhouse gas analysis and forecast system.**

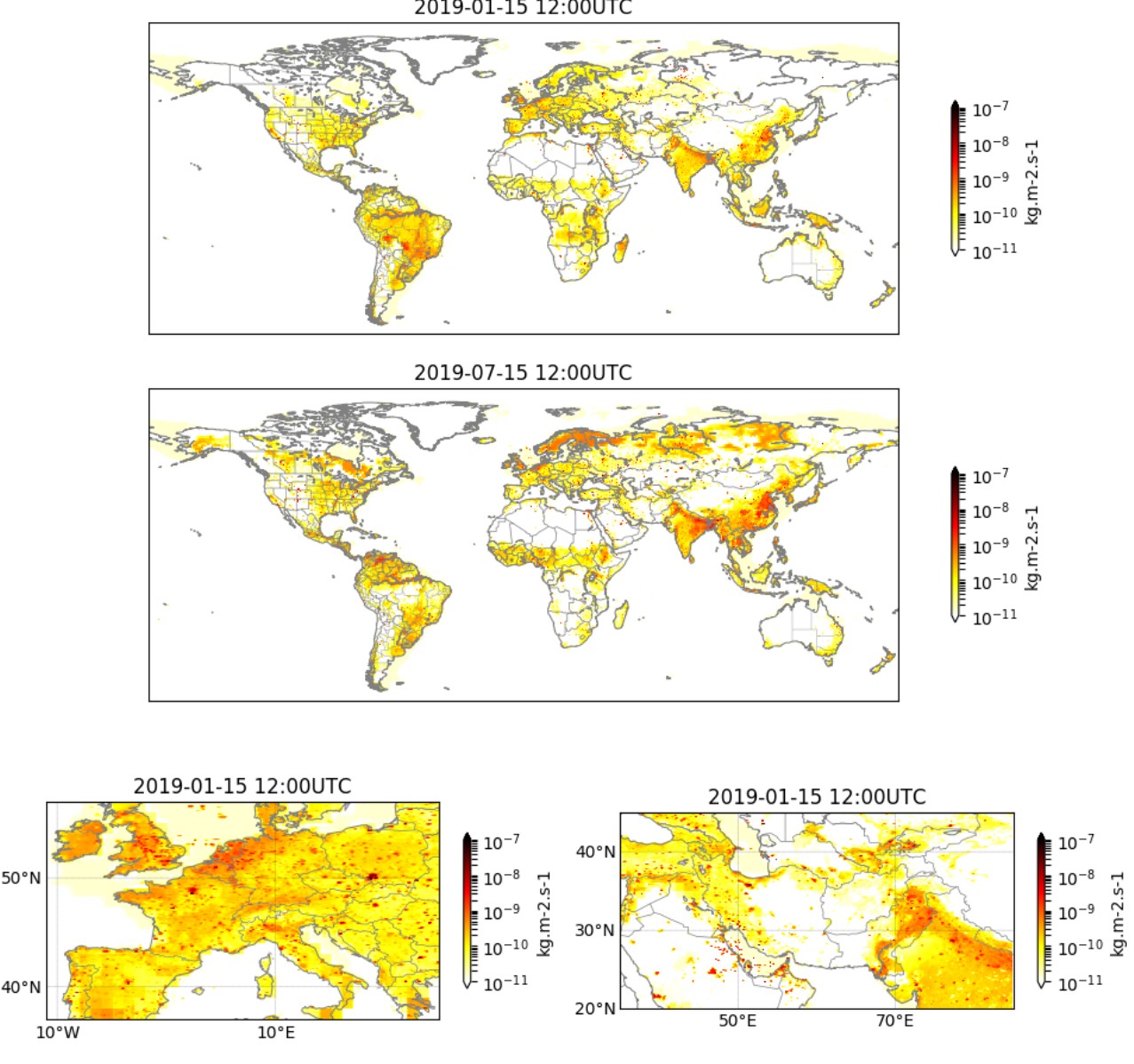

**Figure 4. Examples of combined net fluxes (positive only shown due to the logarithmic scale) that constitute the surface boundary conditions of the IFS high resolution CH₄ forecast. Global and regional scale examples for 2019-01-15 and 2019-07-05 at 12:00 UTC.**

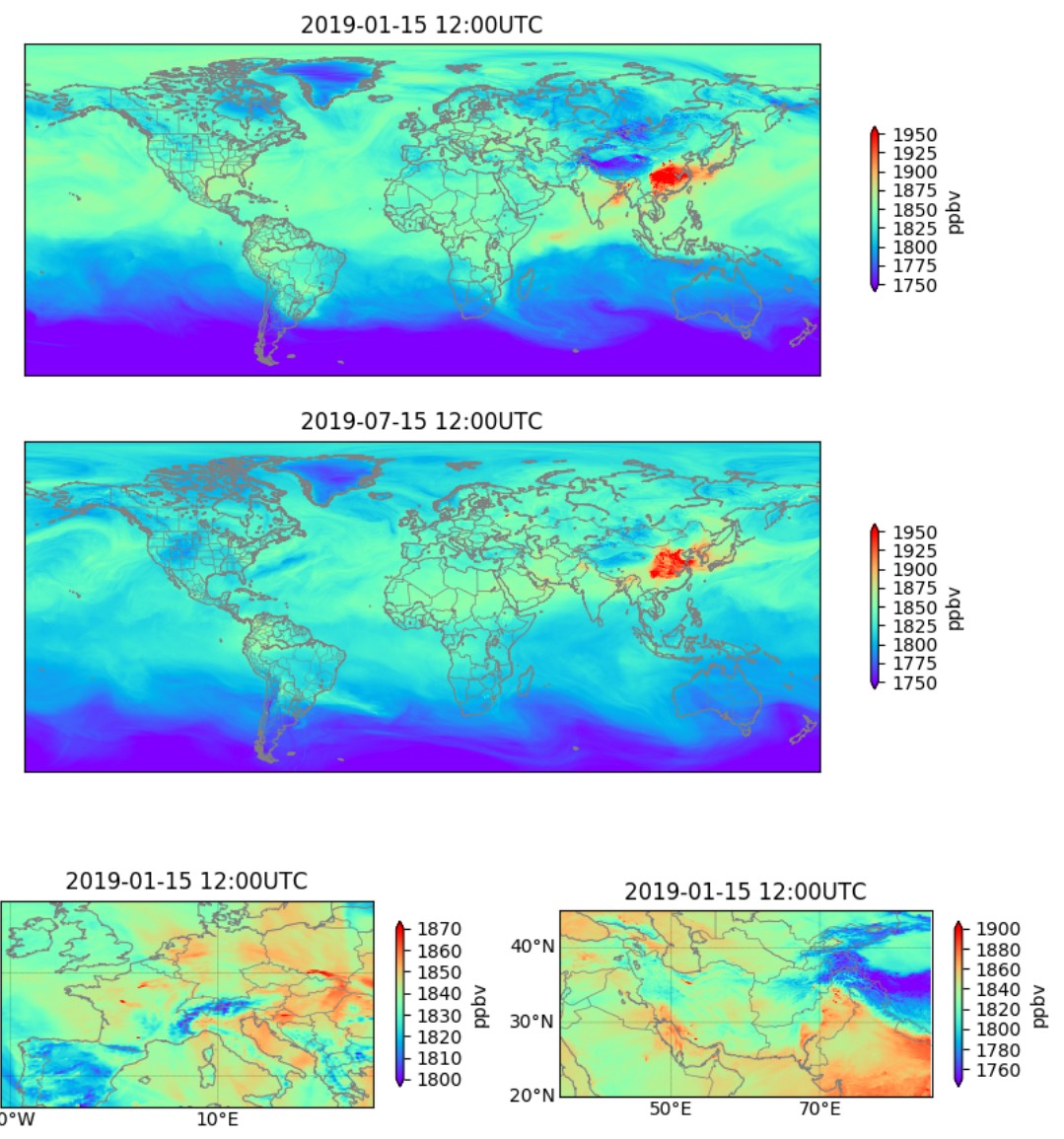


**Figure 5. Examples of outputs of the IFS high resolution CH$_4$ forecasts displaying snapshots at global and regional scale of the total column mean molar fractions for 2019-01-15 and 2019-07-15 at 12:00 UTC. Lower panels show parts of Europe (left) and Middle East (right).**

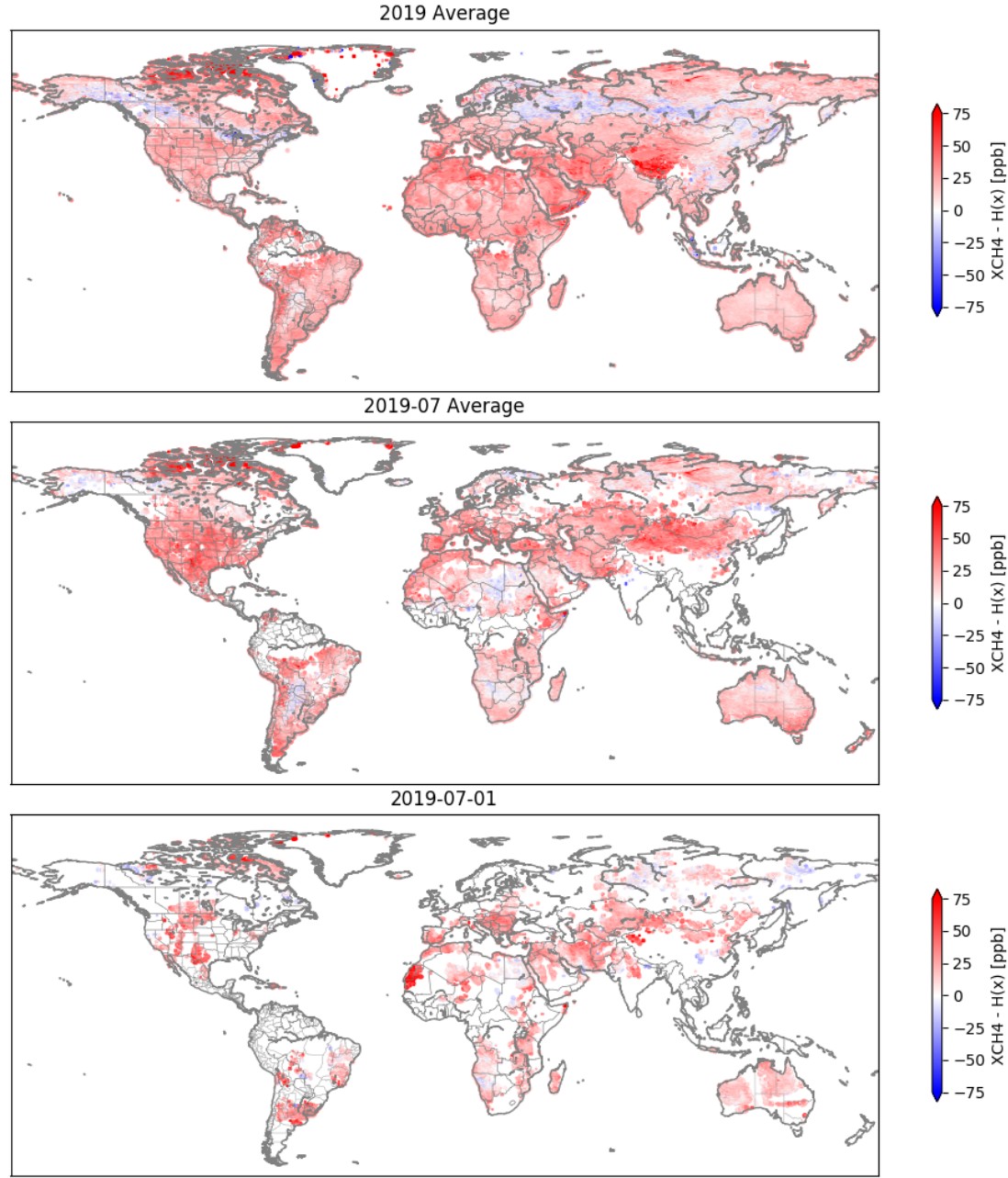


**Figure 6. Departures values computed with the observation displayed in figure 1, for the full year 2019, July 2019 and July 1st, 2019.**

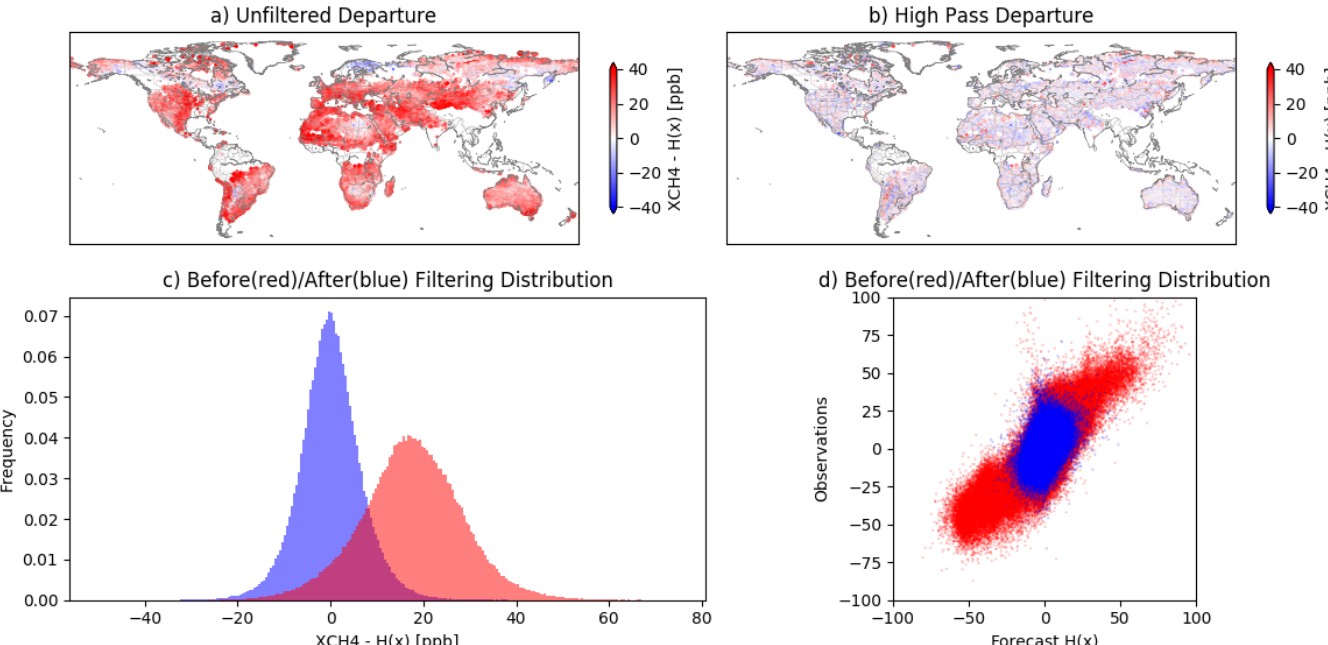

**Figure 7. Example of the high pass filtering effect over a 30-day window ending 2019-07-01 with a 2° Gaussian kernel length scale. a) The unfiltered departures, b) the filtered departures, c) histograms comparing unfiltered (red) versus the filtered (blue) departures and d) 2D distributions in the observation and forecast space for unfiltered (red) and filtered (blue) data points. Note that the unfiltered data points have been centred around the mean for this plot to make it comparable to the filtered distribution.**


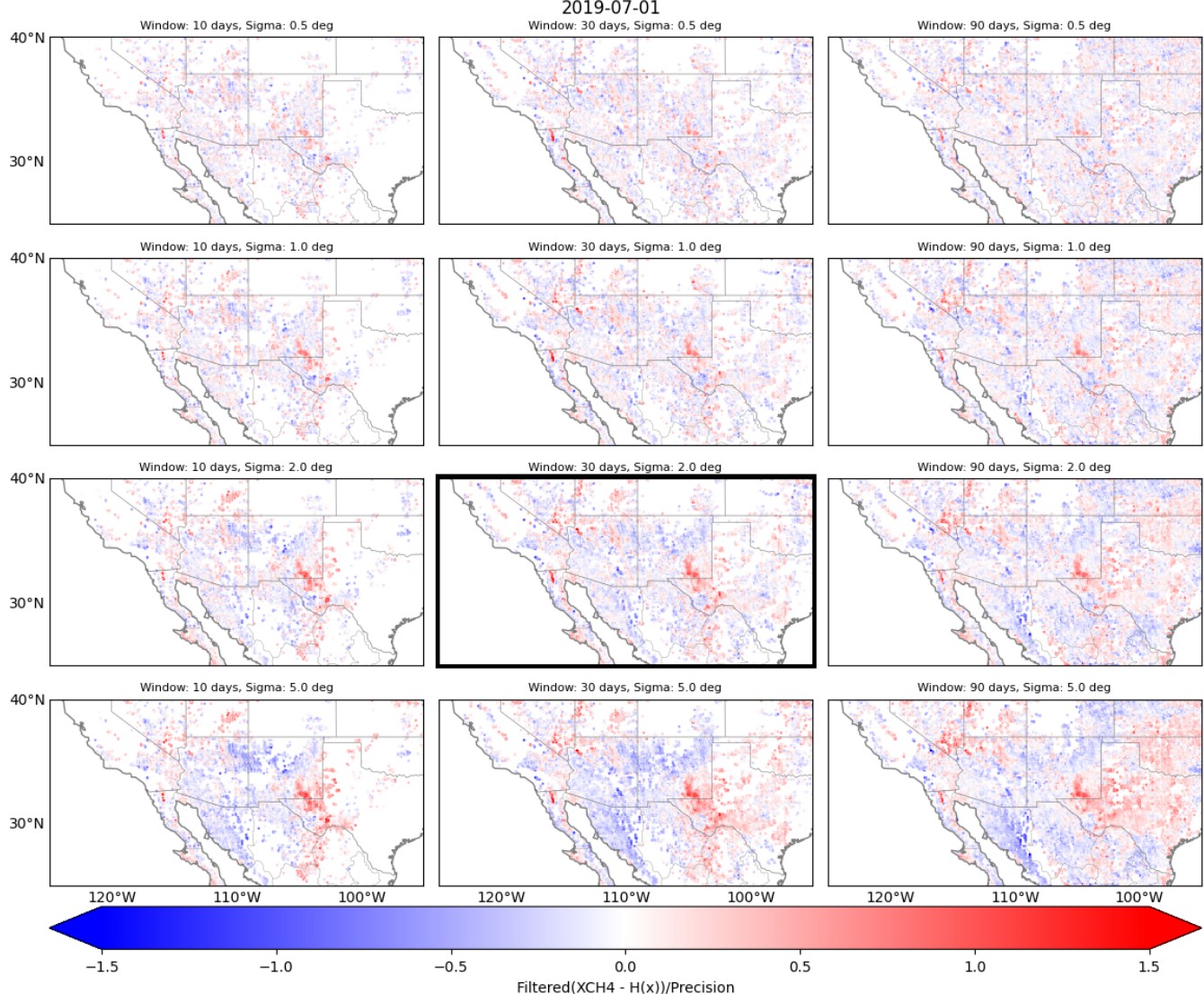

**Figure 8. High pass filtering sensitivity tests on the departures normalized by the instrument precision using 10, 30 and 90 days on the window length (columns) and 0.5, 1.0, 2.0 and 5.0 degrees on the kernel size (rows). The outlined plot shows the selected filter parameter values.**

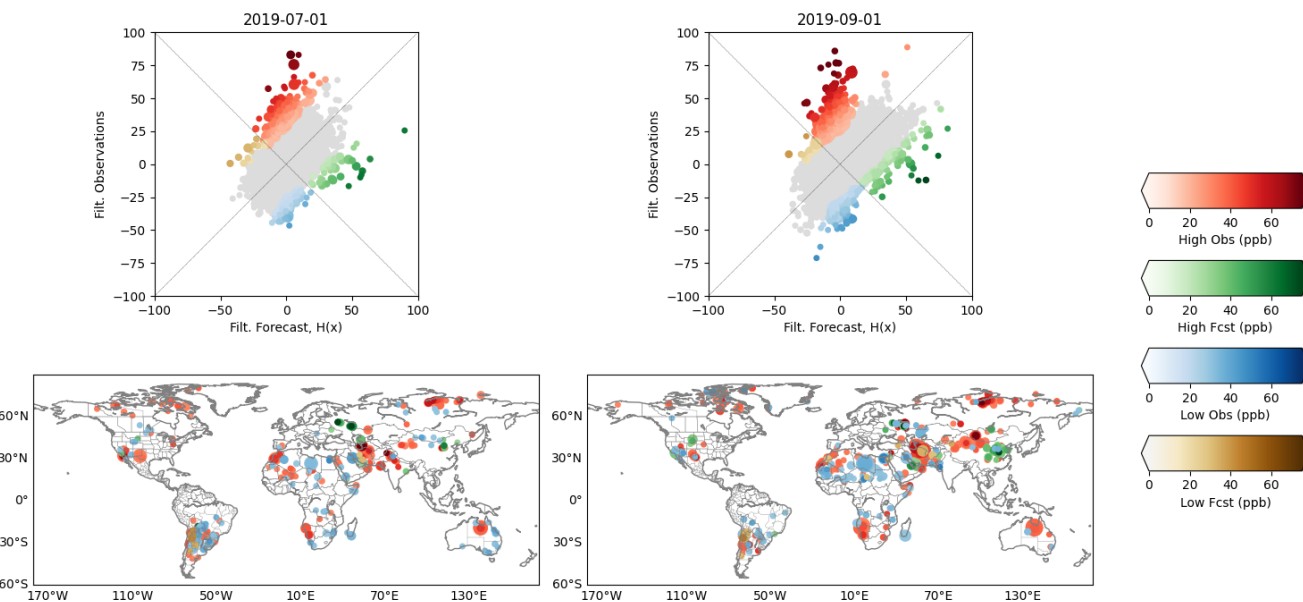


**Figure 9. Examples of the outlier classification. Top panels: global distributions in the observation-first guess space for two different end dates of 30-day window (July 1st 2019 and September 1st 2019). Colours illustrate the four different data classes with number indicating the amount of outliers. Bottom panels: outlier classes localisation example over the globe. Darker dots show larger departures. Larger dots indicate that more occurrences have been detected in the bin and time window.**


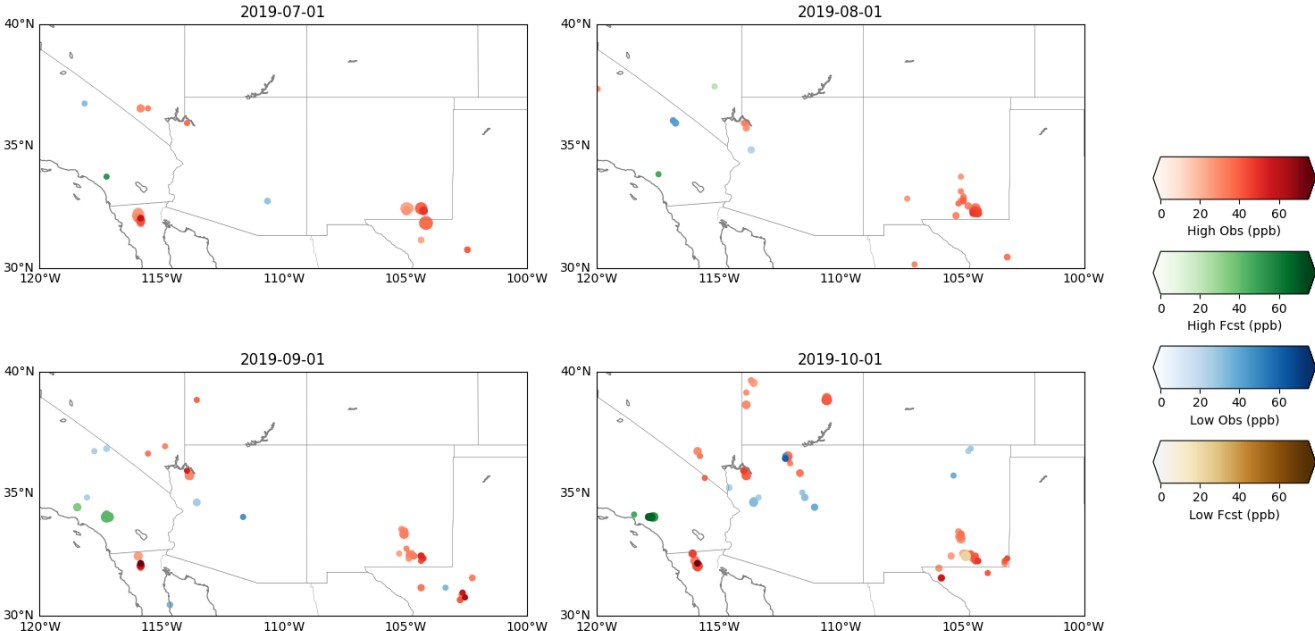

**Figure 10. Outlier detection and classification over south western US region. Dates indicate the end date of the 30-day time window.**


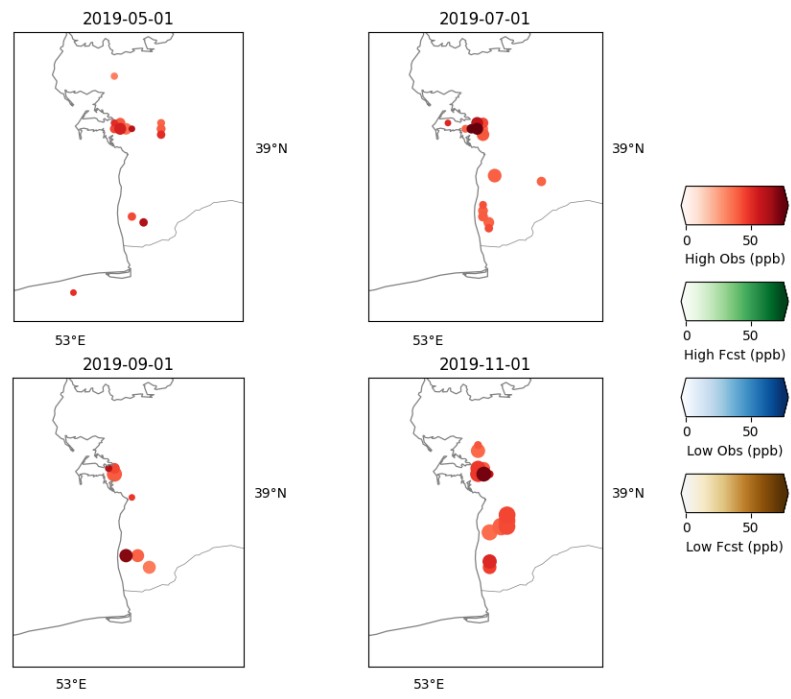

**Figure 11. Outlier detection and classification over Turkmenistan. Dates indicate the end date of the 30-day time window.**

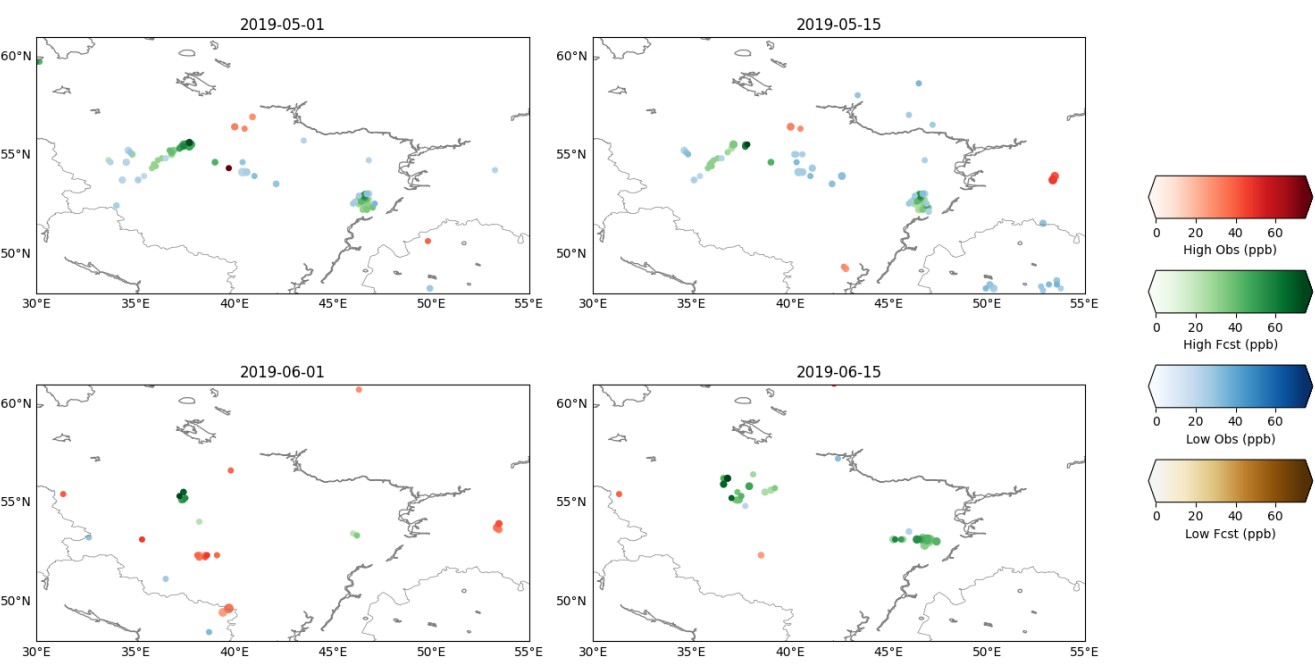

**Figure 12. Outlier detection and classification over western Russia. Dates indicate the end date of the 30-day time window.**

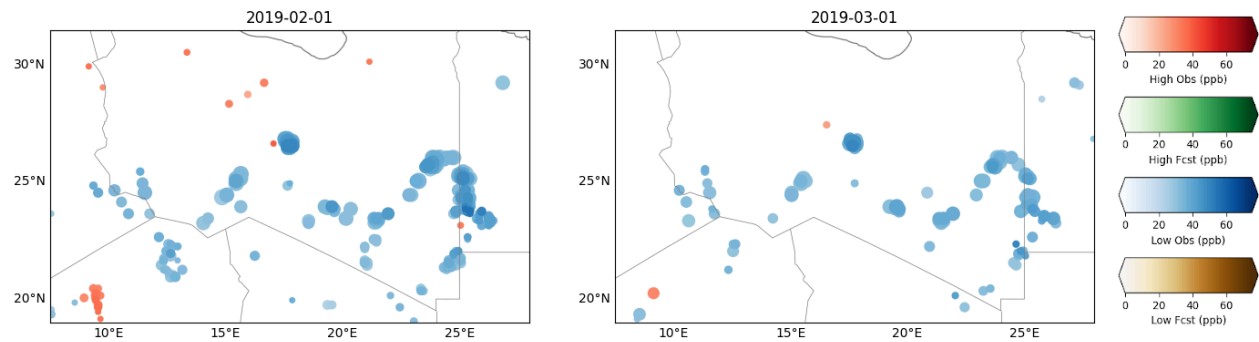

**Figure 13. Outlier detection and classification over Lybia, Egypt and Niger. Dates indicate the end date of the 30-day time window.**


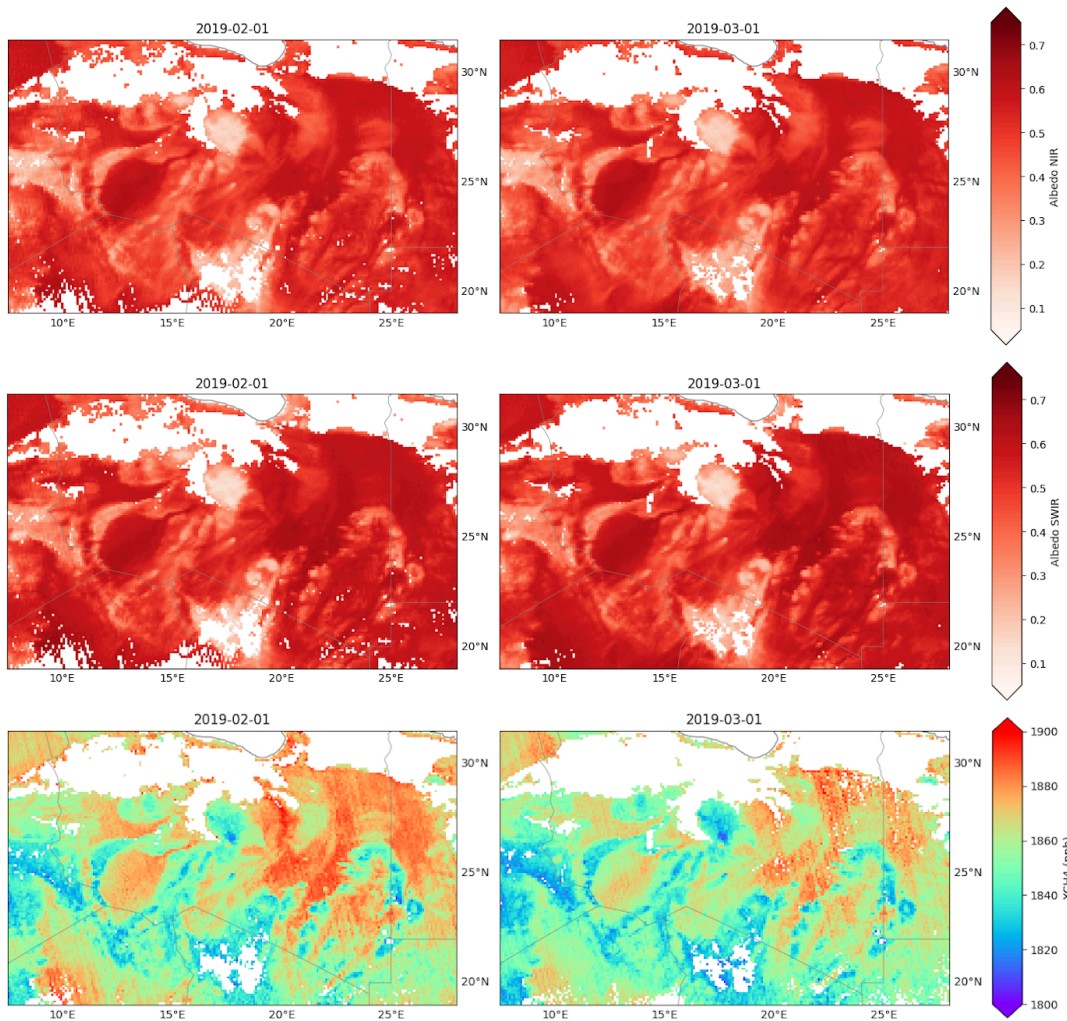

**Figure 14. First row: TROPOMI albedo in the near infrared (NIR) band, second row: TROPOMI albedo in the short-wave infrared (SWIR) band, third row: TROPOMI XCH₄ columns. Maps display averages for the same windows as in Fig. 13.**


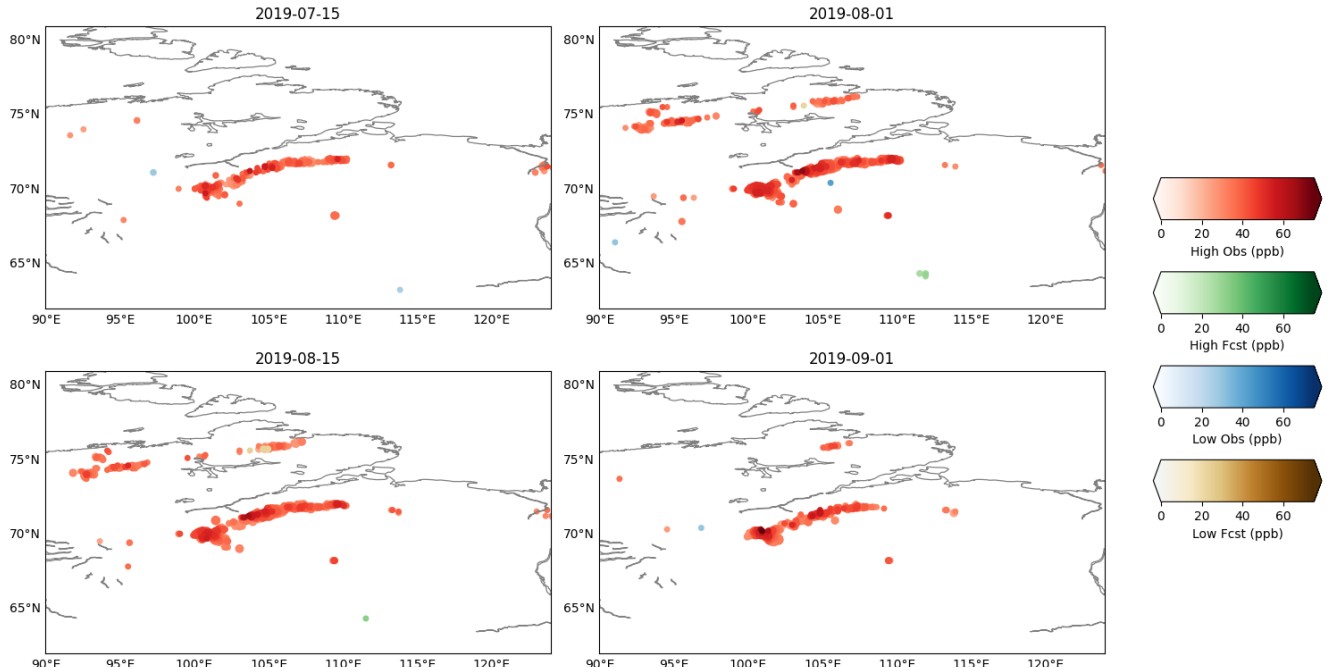

**Figure 15. Outlier detection and classification over Siberia. Dates indicate the end date of the 30-day time window.**

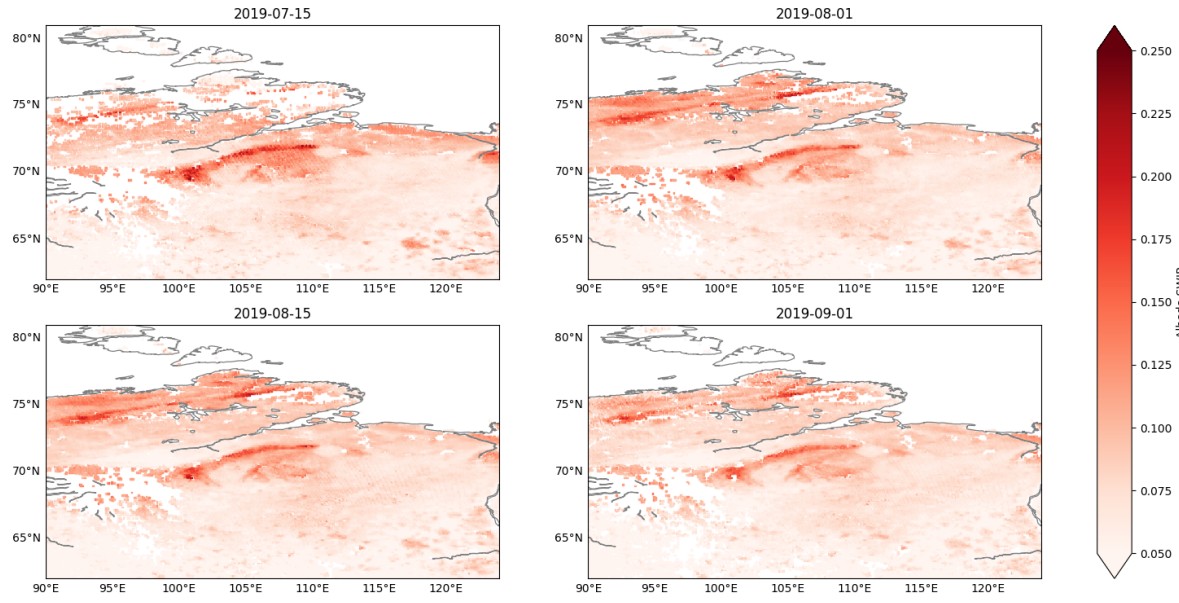

**Fig 16. Albedo in the short-wave infrared (SWIR) band provided by TROPOMI over Siberia. Maps display averages for the same windows as in Fig. 13.**

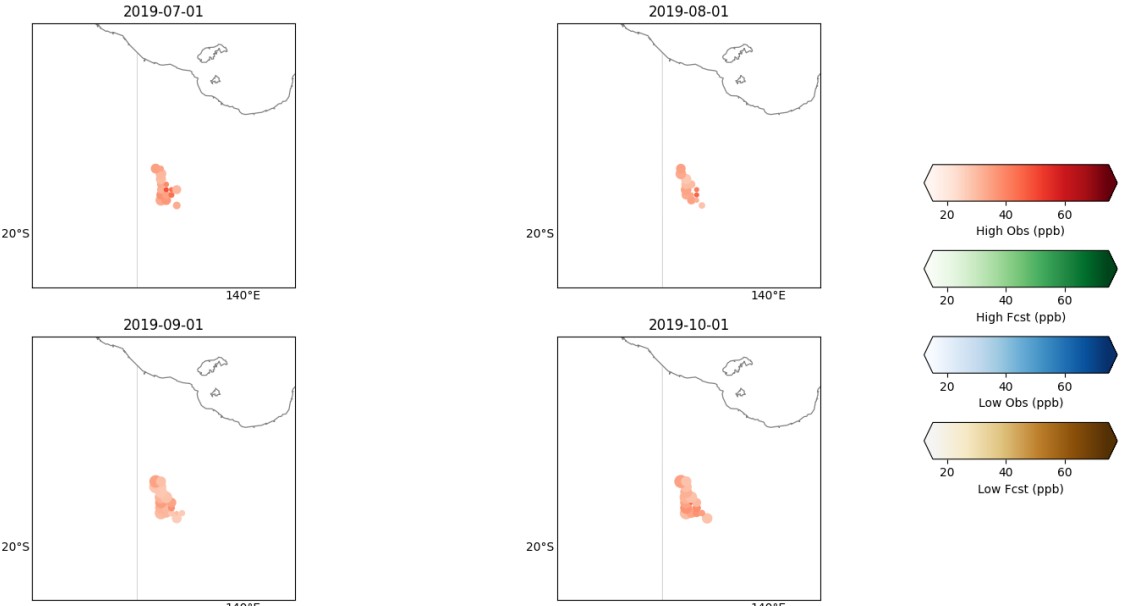

Figure 17. Outlier detection and classification over Australia. Dates indicate the end date of the 30-day time window.

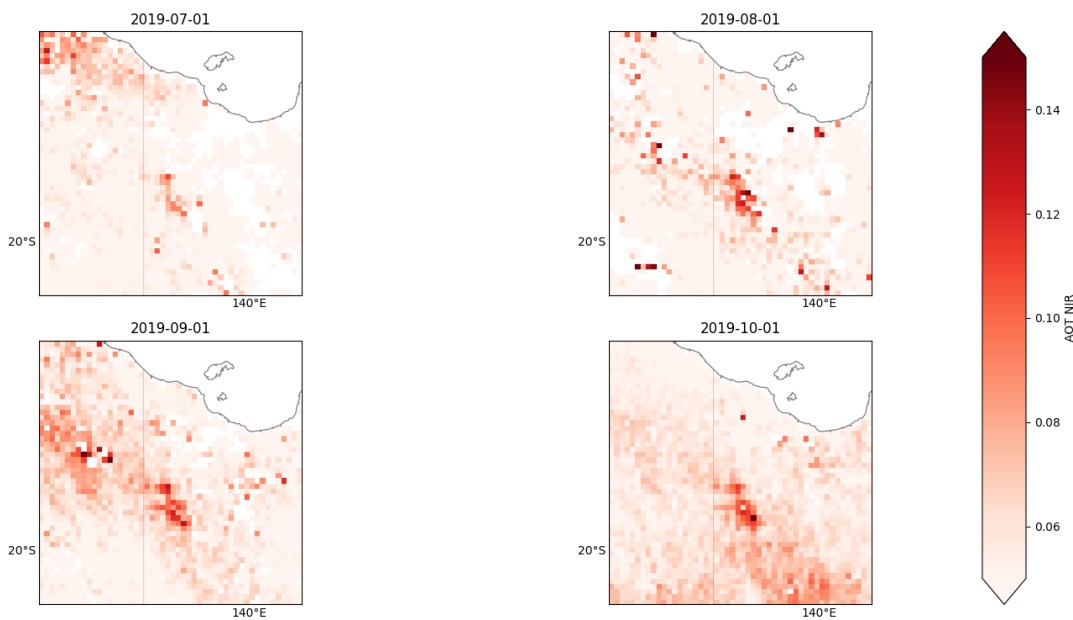

Fig 18. Aerosol optical thickness (AOT) in the near infrared (NIR) band provided by TROPOMI over Australia.
Maps display averages for the same windows as in Fig. 15.