# Peer review of "Systematic detection of local CH4 anomalies combining satellite measurements and high-resolution forecasts."

_Atmospheric Chemistry and Physics, 2020_

## Referee Comment (RC1) · Anonymous Referee #1 · 30 Sep 2020

This manuscript introduces a method to sytematically detect local CH$_4$ anomalies by combining total column CH$_4$ satellite observations from TROPOMI with high resolution CH$_4$ forecasts produced by CAMS. The manuscript is well written but it is at the edge of the scope of ACP, because it rather demonstrates the theoretical potential of the introduced technical method than having general implications for atmospheric science as it is not analysing how well the method can be applied globally (see general comments). However, as the editor has suggested to stay in ACP (instead of moving to AMT), I recommend publication after the following comments have been addressed.

**General Comments**

[Figure]

My main criticism of this manuscript (in the sense of a publication in ACP) is its demonstration character with a lack of general implications for atmospheric science. The advantage is that the method can in principle be applied globally. However, it is not clear if the majority of the detected global anomaly candidates are due to actual unreported or over-reported sources or due to local systematic retrieval biases (e.g. as a consequence of small-scale albedo variations). The presented analysis of the method is limited to a few local case studies (e.g. confirmation of known underreported sources) and does not include an evaluation of the global capabilities to distinguish between missed sources and retrieval biases. Although this would be sufficient for an AMT publication, the global detection statistics and the impact of retrieval biases should be investigated further if possible to better fit the scope of ACP (see also specific comments).

**Specific Comments**

Page 2, Lines 56-59: There are also other relevant studies, e.g. Zhang et al. (2020) or Schneising et al. (2020).

Page 4, Lines 115-117: Is it advisable for the presented method to use a model in which satellite data have already been assimilated (IASI and TANSO)? The assimilated satellite data may already correct for under- or overestimations in the emission data bases to some extent and thus complicate the interpretation.

Page 5, Line 142: Why is Tours visible in Figure 5? What is the origin?

Page 6, Lines 161-163: Please give a reference for the TROPOMI averaging kernel function as a function of pressure.

Equation 6: What does the $lp$ stand for?

Page 8, Line 235: Should "positive" be "negative"?

Page 9, Lines 252-254: You have identified two candidates, which have not been investigated or documented yet: southern Nevada and northern Baja California. You conclude that the latter may be due to local albedo properties as you were not able to

identify a responsible facility. What about the former (Nevada)? Is there are a source or local albedo variations?

Page 10, Line 284-287: "Consistent shapes over months" sounds longer than it actually is. Figure 12 only spans a time period of 6 weeks. The red Turkmenistan features in Figure 10 look similarly consistent over comparable periods. This exemplarily illustrates the difficulty of distinguishing between underreported sources and retrieval biases. Concerning persistent shapes over time, please also discuss the potential impact of temporally and spatially variable small-scale albedo features (e.g. due to snow).

Page 10, Lines 293-295: What is praised as an advantage here (does not only allow for the detection of anomalies but also has the potential of detecting local retrieval errors) is also the main problem: It is not clear if it is possible to reliably distinguish between the two cases in general and there is no global analysis performed to try to approach the answer quantitatively.

Figure 7: Please specify the time period. Is it 2019-07?

Figure 8: Please highlight the plot associated to the final choice of parameters (30 days, 2°) and describe in the caption.

**Technical Corrections**

Please replace CH4 by $CH_4$ in all instances.

Page 7, Line 193: Please delete "and": "... where $d_m$ is the average departure ..."

Page 10, Line 315: Please delete "to": "... emission events could occur ..."

Caption of Figure 4: Is "next fluxes" correct or should it be "net fluxes"?

**References**

Zhang, Y., Gautam, R., Pandey, S., Omara, M., Maasakkers, J. D., Sadavarte, P., Lyon, D., Nesser, H., Sulprizio, M. P., Varon, D. J., Zhang, R., Houweling, S., Zavala-Araiza,

[Figure]

D., Alvarez, R. A., Lorente, A., Hamburg, S. P., Aben, I., and Jacob, D. J.: Quantifying methane emissions from the largest oil-producing basin in the United States from space, Science Advances, 6, https://doi.org/10.1126/sciadv.aaz5120, 2020.

Schneising, O., Buchwitz, M., Reuter, M., Vanselow, S., Bovensmann, H., and Burrows, J. P.: Remote sensing of methane leakage from natural gas and petroleum systems revisited, Atmos. Chem. Phys., 20, 9169-9182, https://doi.org/10.5194/acp-20-9169-2020, 2020.

---

## Referee Comment (RC2) · Anonymous Referee #2 · 8 Nov 2020

The authors have developed a method of combining modeled forecasts and satellite observations of CH4 to isolate local CH4 anomalies that might be associated with discrepancies in anthropogenic emissions. Identifying emissions from point sources such as power plants against background CH4 can be challenging. The manuscript presents a clever approach for doing so that takes advantage of the newly available high-resolution CH4 retrievals from TROPOMI and the equally high-resolution CAMS forecasts. The confirmation of the previously identified CH4 source in western Turkmenistan is a nice demonstration of the potential utility of the approach. I recommend the manuscript for publication in ACP after the authors have addressed my comments below.

[Figure]

Main comments

1) My main concern is that the specification of the filter parameters seems to be subjective. On lines 209-211 the authors explain that specifying a length scale (sigma) of 5 degrees retains large-scale structures in the signal, whereas a length scale of 0.5 degrees results in the loss of too much signal. As a result, they selected a length scale of 2 degrees. How did they decide when too much signal is lost? It would be helpful if the authors could be more quantitative. In addition, what is meant by "large-scale" in this context? For example, the panels in Fig. 8 for sigma of 0.5, 1.0, and 2.0 for both the 10-day and 30-day windows all look similar to me. I can see the three regions of CH4 enhancements in the southwestern US and Mexico in all six panels, so it is unclear why the case for sigma = 2.0 with a 30-day window is best. If there is a tacit assumption being made about what specific scales are of interest and the type of emissions that are associated with those scales, that assumption should be explicitly stated as it places a constraint in the utility of the approach.

2) In a similar vein, on lines 217-221 the authors stated that they selected a three-sigma threshold for the outlier classification because it "provides suitable results." It is unclear how "suitable" is defined here? They stated that a narrower range "starts to fail isolating important anomalies and conversely a wider range might fail to capture useful information." What are the important anomalies that are not detected with the narrower range? The discussion here should be expanded to give the reader a better sense of what are the implications of this threshold for the type of anomalies that can be detected.

3) I am also concerned about the lack of discussion about the potential impact of biases. The filtering does remove "large-scale" biases, but it would be helpful if the authors could comment on the impact of spatially and temporally varying biases on smaller scales (i.e., scales between sigma and the "large-scale"). These could arise from the influence of transport errors, for example.

4) Line 269-270: Is the over-prediction in the Los Angeles area seen only in the Aug-Sept period (i.e., in the right panel)? It doesn't seem to be present in the Jun-Jul plot (left panel). If that is the case, it should be noted in the manuscript. What could be the cause of this temporal variation?

Technical comments

1) Lines 14, 81, 108, 109, 114, 162, 170, 275: A space is needed between the numbers and units, e.g., "9 km" instead of "9km".

2) Line 52: I believe the "Pandley et al." reference should be "Pandey et al."

3) Line 75: Please change "earth" to "Earth".

4) 4) Lines 336, 341, 343, 354, 360: The line spacing between the references is irregular.

5) Figure 5: It is difficult to see the features in the plot for Europe. Since the objective here is to show that ability of the model to capture fine scale features, why not plot the European sector with a different scale to better emphasize these features?

6) Figure 7: It is unclear what is the time period for the data shown here.

7) Figure 9: It is difficult to see the yellow/gold colors for the low forecast category.

---

## Author Comment (AC1) · 30 Nov 2020

**General Comments**

My main criticism of this manuscript (in the sense of a publication in ACP) is its demonstration character with a lack of general implications for atmospheric science. The advantage is that the method can in principle be applied globally. However, it is not clear if the majority of the detected global anomaly candidates are due to actual unreported or over-reported sources or due to local systematic retrieval biases (e.g. as a consequence of small-scale albedo variations). The presented analysis of the method is limited to a few local case studies (e.g. confirmation of known underreported sources) and does not include an evaluation of the global capabilities to distinguish between missed sources and retrieval biases. Although this would be sufficient for an AMT publication, the global detection statistics and the impact of retrieval biases should be investigated further if possible to better fit the scope of ACP (see also specific comments).

*We firstly thank the reviewer with the review that led to significantly improve the paper. We agree with the reviewer and the editor that the article is at the edge of the ACP scope. The goal of the article is to present the capability we newly developed using part the ECMWF IFS data assimilation system to perform this automatic detection so we were not sure if this would fit the AMT journal scope as well. We are happy to have the article moved to AMT if the editor and the reviewer think that this is more appropriate.*
*We showcase few cases studies (now 5) to show the potential of such methods but also highlight the short comings. Going into details on more case studies to provide a global analysis would significantly lengthen and change the paper. We have however added one more case study to strengthen the point on local persistent biases in response to one specific comment below (see section 4.3 and figure 14).*
*Significant restructuration of the paper would be needed with providing a global evaluation of the capabilities and this would require diagnosing the system for an extended time period (probably several years) to provide a robust analysis. This will be done in further steps as we plan to have this system running routinely soon. But this cannot be done within the time frame of this present review. We do however have changed figures to include the global detection results for few months (now in figure 9) with numbers to show how the system is performing globally. We also now mention this point in the conclusion motivating for a further global analysis once the data set will be available.*

**Specific Comments**

Page 2, Lines 56-59: There are also other relevant studies, e.g. Zhang et al. (2020) or Schneising et al. (2020).

*We have added the references to the text.*

Page 4, Lines 115-117: Is it advisable for the presented method to use a model in which satellite data have already been assimilated (IASI and TANSO)? The assimilated satellite data may already correct for under- or overestimations in the emission data bases to some extent and thus complicate the interpretation.

*Our current data assimilation technique does not correct the emissions but only the concentrations. We clarify the text to make this point clearer.*

Page 5, Line 142: Why is Tours visible in Figure 5? What is the origin?

*We have added a brief explanation in the text.*

Page 6, Lines 161-163: Please give a reference for the TROPOMI averaging kernel function as a function of pressure.

*We have now added the reference in the text.*

Equation 6: What does the lp stand for?

*We subscript lp here was wrong as it should stand for high pass. We changed the subscript to hp and clarified the text accordingly.*

Page 8, Line 235: Should "positive" be "negative"?

*This has been corrected.*

Page 9, Lines 252-254: You have identified two candidates, which have not been investigated or documented yet: southern Nevada and northern Baja California. You conclude that the latter may be due to local albedo properties as you were not able to identify a responsible facility. What about the former (Nevada)? Is there are a source or local albedo variations?

*This is likely the case as well for the enhancement seen at the Nevada-Arizona border. We clarified the text accordingly.*

Page 10, Line 284-287: "Consistent shapes over months" sounds longer than it actually is. Figure 12 only spans a time period of 6 weeks. The red Turkmenistan features in Figure 10 look similarly consistent over comparable periods. This exemplarily illustrates the difficulty of distinguishing between underreported sources and retrieval biases. Concerning persistent shapes over time, please also discuss the potential impact of temporally and spatially variable small-scale albedo features (e.g. due to snow).

*We now have improved the Turkmenistan figure to a more zoomed version to show that the shape of the feature is not consistent over months like in the Siberian case. Regarding the Siberian case we can only show it over a reduced span (figure 13 spans over more than 10 weeks as the window length is 30 days i.e. around 4 weeks) as the measurements are only available for a short summer period over these latitudes. To make our point stronger and clearer we have identified an additional case over Australia illustrated with figure 14 were four 30-day windows are displayed over 4 months. We also address the potential impact of temporally and spatially variable small-scale albedo features (e.g. due to snow). We clarified the text accordingly.*

Page 10, Lines 293-295: What is praised as an advantage here (does not only allow for the detection of anomalies but also has the potential of detecting local retrieval errors) is also the main problem: It is not clear if it is possible to reliably distinguish between the two cases in general and there is no global analysis performed to try to approach the answer quantitatively.

*The response to the comment above partially answers this comment. We do agree with the reviewer of the current limitation and that this issue deserves further works as already mentioned at the end of section 4.3. We now have clarified the text accordingly.*

Figure 7: Please specify the time period. Is it 2019-07?

*This has been clarified.*

Figure 8: Please highlight the plot associated to the final choice of parameters (30 days, 2_) and describe in the caption.

*This has been clarified.*

**Technical Corrections**

Please replace CH4 by $CH_4$ in all instances.

*Fixed*

Page 7, Line 193: Please delete "and": "... where $d_m$ is the average departure ..."

*Fixed*

Page 10, Line 315: Please delete "to": "... emission events could occur ..."

*Fixed*

Caption of Figure 4: Is "next fluxes" correct or should it be "net fluxes"?

*Fixed*

---

## Author Comment (AC2) · 30 Nov 2020

The authors have developed a method of combining modeled forecasts and satellite observations of CH4 to isolate local CH4 anomalies that might be associated with discrepancies in anthropogenic emissions. Identifying emissions from point sources such as power plants against background CH4 can be challenging. The manuscript presents a clever approach for doing so that takes advantage of the newly available high-resolution CH4 retrievals from TROPOMI and the equally high-resolution CAMS forecasts. The confirmation of the previously identified CH4 source in western Turkmenistan is a nice demonstration of the potential utility of the approach. I recommend the manuscript for publication in ACP after the authors have addressed my comments below.

**Main comments**

1) My main concern is that the specification of the filter parameters seems to be subjective. On lines 209-211 the authors explain that specifying a length scale (sigma) of 5 degrees retains large-scale structures in the signal, whereas a length scale of 0.5 degrees results in the loss of too much signal. As a result, they selected a length scale of 2 degrees. How did they decide when too much signal is lost? It would be helpful if the authors could be more quantitative. In addition, what is meant by "large-scale" in this context? For example, the panels in Fig. 8 for sigma of 0.5, 1.0, and 2.0 for both the 10-day and 30-day windows all look similar to me. I can see the three regions of CH4 enhancements in the southwestern US and Mexico in all six panels, so it is unclear why the case for sigma = 2.0 with a 30-day window is best. If there is a tacit assumption being made about what specific scales are of interest and the type of emissions that are associated with those scales, that assumption should be explicitly stated as it places a constraint in the utility of the approach.

*We thank the reviewer for this helpful comment. We agree that the summitted version of the article do not discuss quantitatively the selection of the filter parameters. We do not have however found a quantitative metric or criteria to find the optimal parameters values for the size of the convolution kernel and the length of the window. Given the answer to the comment 2) and the change to an outlier threshold now based on the measurement precision, we justify the size of the kernel as follows. If the kernel is too small, the filtered signal is getting weaker than the measurement precision and very few to no detections of anomalies is made. Conversely if the kernel is large the signal is strong but to the risk of picking up larger patterns than the targeted features, i.e. which are directly related to local emissions in the $CH_4$ atmospheric distribution. Regarding the window length, we attempted to explain in the text that a short window could fail to give enough coverage to correctly run the convolution filter. In the opposite a long window will maximise the chances to have to good coverage, but the method would lose its ability to provide temporal variability. For those reasons we chose to run the filtering with a 2 degrees kernel size over a window of 30 days. We have now detailed and clarified the text accordingly.*

2) In a similar vein, on lines 217-221 the authors stated that they selected a three sigma threshold for the outlier classification because it "provides suitable results." It is unclear how "suitable" is defined here? They stated that a narrower range "starts to fail isolating important anomalies and conversely a wider range might fail to capture useful information." What are the important anomalies that are not detected with the narrower range? The discussion here should be expanded to give the reader a better sense of what are the implications of this threshold for the type of anomalies that can be detected.

*We thank the reviewer for this very helpful comment. Concerning the three-sigma standard deviation threshold, we have chosen this value because it is what is commonly used to account for outliers. We however agree that this is not an objective criterion. We have then updated and improved the outlier classification and is now based on the measurement precision that is provided with the TROPOMI $CH_4$ product. When the absolute value of the filtered departure is above the precision value, we consider this as an anomaly to be displayed. We clarified and modified the text and figures of the paper accordingly.*

3) I am also concerned about the lack of discussion about the potential impact of biases. The filtering does remove "large-scale" biases, but it would be helpful if the authors could comment on the impact of spatially and temporally varying biases on smaller scales (i.e., scales between sigma and the "large-scale"). These could arise from the influence of transport errors, for example.

*We are not sure to fully understand the reviewer's concern. Features of scales above sigma (the kernel length scale) are removed by the filter. Thus the 'scales between sigma and the "large-scale"' are not impacting the results and analysis here. We clarified the text accordingly (at the end of section 2.1) by defining large-scale in the paper's*

*context of being the combination of synoptic-scales (2000 km or more) and meso-alpha-scale (between 200 km and 2000 km).*

4) Line 269-270: Is the over-prediction in the Los Angeles area seen only in the Aug-Sept period (i.e., in the right panel)? It doesn't seem to be present in the Jun-Jul plot (left panel). If that is the case, it should be noted in the manuscript. What could be the cause of this temporal variation?

*We have now detailed the section 4.2 to take into account the reviewer's comment. We have added an additional figure (now figure 10) that covers four 30 days window over the southwestern US and northern Mexico area.*

**Technical comments**

1) Lines 14, 81, 108, 109, 114, 162, 170, 275: A space is needed between the numbers and units, e.g., "9 km" instead of "9km".

*Fixed*

2) Line 52: I believe the "Pandley et al." reference should be "Pandey et al."

*Fixed*

3) Line 75: Please change "earth" to "Earth".

*Fixed*

4) 4) Lines 336, 341, 343, 354, 360: The line spacing between the references is irregular.

*Fixed*

5) Figure 5: It is difficult to see the features in the plot for Europe. Since the objective here is to show that ability of the model to capture fine scale features, why not plot the European sector with a different scale to better emphasize these features?

*Fixed. We adjusted the color scale.*

6) Figure 7: It is unclear what is the time period for the data shown here.

*Fixed. We added the end of window date in the figure caption.*

7) Figure 9: It is difficult to see the yellow/gold colors for the low forecast category.

*Fixed, we have changed it to darker color scale.*

---

## Author Response (AR2)

*Dear Referee and Editor,*

*We are thankful for these helpful comments, that helped to improve the quality of the paper and the future implementation of this system. We hope that the main point about the retrieval biases is now addressed correctly as we do now clearly acknowledge the current limitation of the method. See our response to the comments in italic font below.*

**Referee #1 comments:**

I appreciate the further analyses as well as the additional global view in Figure 9 and also understand that it is hardly possible to carry out a complete and comprehensive analysis of the capabilities and short-comings of the method within the framework of this paper. Therefore, I think that a publication in AMT would be more appropriate to present an introduction of a new interesting technical method but I do not insist on moving the article if the editor has a different opinion.

However, without a more detailed evaluation of the global capabilities some of the statements and conclusions have to be weakened (see below). I would also propose to emphasise more clearly what is actually done already in the title and/or in the abstract: local XCH4 anomalies are detected (by combining satellite measurements and forecasts), which are due either to actual emission anomalies relative to the CAMS emissions or systematic biases in the satellite data or a combination of both.

*We have modified the abstract and the title accordingly*

The distinction between the two cases has to be made by (subjective) interpretation and is prone to error. I do not think that the decision should be made by solely assessing the persistence of features as suggested in the manuscript. Persistent emissions can also lead to persistent features as there is not always a clear plume structure in satellite GHG data, in particular for non-point sources, when topography causes accumulation, and when using a 30-day time window (see also specific comments). How persistent (in space and time) is a pattern allowed to be to be considered real? You need additional information, e.g. to check if the features are correlated with albedo features (like you do for Figure 13), to classify patterns as retrieval biases. In principle, you also have to check the albedo features in the cases where anomalies are expected (e.g. Permian and Turkmenistan in Figures 10 and 11) to avoid expectation bias. On the other hand, emission patterns could indeed be correlated with albedo (e.g. wetlands or facilities vs. surroundings) complicating the interpretation. Please elaborate more on these issues and discuss them in a more balanced way.

For example, in case of the new example shown in Figure 14, where there is no correlation of the outlier pattern with albedo, I would be cautious to classify this feature as retrieval bias unless another good explanation for a potential retrieval bias has been found.

*We agree with the reviewer about the limitations of the method and we have clearly stated them in the abstract. Section 4.3 and the conclusion. We also now have substantially modified and complemented section 4.3 and the associated figures to provide clearer and more accurate discussion about the identification of the retrieval errors.*

Specific Comments

Page 2, Lines 58-59: The sentence is a little misleading because the cited papers analyse different regions, but all include the Permian basin. Therefore, I suggest to change it to something like: "... large and extended enhancements in different US oil and gas production regions such as the Permian basin."

*The sentence has been adjusted as suggested.*

Page 3, Lines 86-87: What is the difference between instrument precision and random error?

*We have clarified the statement*

Page 4, Lines 117-121: Your data assimilation technique only corrects the concentrations and not the emissions.

But this is exactly the potential problem, isn't it? As a consequence, H(x) (in ppb) potentially depends on patterns observed by IASI and TANSO. Or am I getting something wrong here? Assume there is a (unknown) source, which is observed by TANSO and TROPOMI. Then the concentrations in the forecast are corrected upwards due to TANSO and the difference d to TROPOMI (which also sees an enhancement) is getting smaller because of the assimilation. The other way round, isn't it possible that a potential bias in the IASI or TANSO data, which is assimilated, causes an artificial outlier of your method although emission data bases are actually consistent with the TROPOMI measurements? Along these lines, wouldn't it be better to use a model without assimilation of satellite data as starting point if you want to assess the quality of emission data bases?

*The assimilation is having an expected minimal impact for correcting close to sources concentrations when the high-resolution departures are computed for two main reasons:*
   1. *The assimilation impact close to the surface is weak due to the poor coverage of TANSO and the low sensitivity of IASI close to the surface. See Massart et al., 2014 for details.*
   2. *As described in section 2.2 and figure the assimilation is acting at lower resolution (around 25km) at least 4 days before the high-resolution departures are computed.*
*This is the way our operational 9km high-resolution ECMWF forecasts are initialized operationally, and we take advantage of this to perform this demonstration of monitoring in a cheap and efficient way. We agree that running another high-resolution suite of runs free of CH4 data assimilation would make things strictly cleaner but more costly for expected very minimal differences close to the surface.*

*We have now added clarifications at the end of the paragraph.*

Page 6, Lines 166-168: Is the averaging kernel function as a function of pressure really discussed in the cited paper? Moreover, the paper analyses a different algorithm than the one used here. Please cite a paper describing the averaging kernels of the operational TROPOMI algorithm if possible.

*We now put the correct reference (using Hu et al., 2016).*

Page 9, Lines 265-267: Please check if there is actually a correlation with surface albedo features (as in the case of Figure 13).

*We didn't find clear evidence of albedo features corresponding to the enhancements. We also do not identify facilities that could be the reason of those enhancements either. For the northern Baja California we have identified that the feature could be correlated with scattering parameters see figures below. We have then changed the statement accordingly.*

[Figure]

***Aerosol optical thickness (AOT) in the short-wave infrared band (SWIR) values provided with the TROPOMI CH4 retrievals. Maps provides averages corresponding to the windows used in figure 10.***

[Figure]

*Aerosol optical thickness (AOT) in the near infrared band (NIR) values provided with the TROPOMI CH4 retrievals. Maps provides averages corresponding to the windows used in figure 10.*

Page 10, Lines 301-303: This statement is too strong. Please write e.g. "potential retrieval error artefacts". Persistence isn't everything because plumes are not always visible in daily GHG data and may disappear when using multi-day time windows if the wind direction changes. As a consequence, it is possible that you only get an anomaly right above the source with your method (see also general comments). Please revise this section accordingly.

*We have clarified the statement further. We however still want to emphasize that such a very similar shape seen an extended period of time as in Fig. 13 is extremely unlikely to be an emission signature. Section 4.3 has been re-written, figures amended with additional plots displaying albedo and scattering features retrieved from TROPOMI to complement the discussion.*

Page 10, Lines 305-307: I would be cautious to classify this feature as retrieval bias when there is no correlation with albedo features. Are there other potential explanations? (Other features causing biases? Could it be a real signal?)

*Section 4.3 has been re-written, figures amended with additional plots displaying albedo and scattering features retrieved from TROPOMI to make the point clearer. The match for the Siberian feature is quite striking.*

Page 11, Lines 317-318: Please add a sentence that the distinction between over-/under-reported sources and local retrieval errors is challenging and needs correlation analyses with external data sets such as albedo.

*We have added the required sentence.*

Figures 9-14: The colours of the four categories are sometimes hard to distinguish in the maps (in particular with the updated colours in the revised version). Please consider to use different colours or to code the classes additionally in a different way (for example by different symbol shapes or hatching).

*The colours have been changed already as request by reviewer #2 in the first round of review where gold colours have been made darker. We are not sure to what more differentiable colour between red,green,blue and yellow (gold) we could possibly make the scatter plot with. We are also not sure that different shapes or hatching will make things clearer as the number of data points are high. We then increase the size of the dots and make the colours less faint in the scatter plots to improve the readability.*

**Editor comments:**

1) I'm not a modeller which is probably why the discussion of the monitoring suite and the departure is confusing to me:

a. What do you mean by "trajectory" – for me, that is a Lagrangian calculation of the track of an air mass but here it seems to have another meaning

*Trajectory in the variational data assimilation sense is a model forecast within the data assimilation window (here 12 hours) in order to compute the observation departures. Which is different from the model forecasts as this is a model run of few days initialized from the data assimilation analyses. We clarified the text accordingly.*

b. What is the difference between what is done in the monitoring suite and what I would have naively done, namely comparing the model forecast for the time of measurement with the measurement after applying the measurement operator to the model profile?

*There is no difference. We use part of the data assimilation system to perform this monitoring at the exact model time step.*

c. What is the time step of the monitoring suite?

*It is 450 seconds; we now specify this in the text.*

2) I do not understand your outlier classification:
a. What do you mean by "positive filtered observations"? Do you mean filtered observations, which are positive? If so, then the description of the green and golden classes does not match what I see in Figure 9.

*We have clarified the description of the classes.*

b. What is the difference between first-guess and forecast? If there is none, please just use "forecast"

*We have now changed first-guess to forecast*

c. Why is the green category representative of "over-reported or under-reported plumes" as stated in line 256?

*We have corrected the sentence.*

d. I have trouble seeing the benefit of separating into four instead of just two groups. It would be good to provide for each of these categories an example of an application to illustrate its usefulness.

*Separating into only two categories, I assume positive and negative departures only, will not allow to disentangle when anomalies are due to observations or due to the forecasts.*

*We also now provide example using the low observation category (blue), illustrating its usefulness. For the low forecast category (gold) the number of occurrences is very low and very sparse and probably not significant in our current monitoring dataset we then decide not to illustrate it in our paper. This category could be however important to identify low model/inventory biases if they had to occur.*

3) As also pointed out by the reviewer, the use of this method to evaluate emission inventories is undermined by the assimilation of other satellite data. Comparison to a control run would be a cleaner way of achieving this goal.

*Please see the response to the reviewer's comment.*

4) I also agree with the reviewer that this type of comparison is useful to identify retrieval artefacts but it is not straight forward as it is difficult to disentangle differences coming from deficiencies in the emission inventories and modelling set-up from retrieval problems.

*Please see the response to the corresponding reviewer comments. We have now detailed the text accordingly to take into account this important point.*